# Learning to Solve Complex Problems
# via Dataset Decomposition

**Wanru Zhao**[a,b]      **Lucas Caccia**[b]      **Zhengyan Shi**[b]      **Minseon Kim**[b]

**Weijia Xu**[b]                    **Alessandro Sordoni**[b,c]

[a]University of Cambridge    [b]Microsoft Research    [c]Mila - Quebec AI Institute

## Abstract

Curriculum learning is a class of training strategies that organizes the data being exposed to a model by difficulty, gradually from simpler to more complex examples. This research explores a reverse curriculum generation approach that recursively decomposes complex datasets into simpler, more learnable components. We propose a teacher-student framework where the teacher is equipped with the ability to reason step-by-step, which is used to recursively generate easier versions of examples, enabling the student model to progressively master difficult tasks. We propose a novel scoring system to measure data difficulty based on its structural complexity and conceptual depth, allowing curriculum construction over decomposed data. Experiments on math datasets (MATH and AIME) and code generation datasets demonstrate that models trained with curricula generated by our approach exhibit superior performance compared to standard training on original datasets.

## 1   Introduction

When teaching language models (LMs) to solve mathematical problems, one common solution is to apply supervised learning on a dataset of problems with worked-out solution steps. This is, to some extent, in stark contrast to how humans are educated and generally taught to solve problems, e.g. with a *curriculum*, where a teacher presents simpler concepts first, to build a foundation for later tackling more challenging tasks.

Transposing this strategy to train machine learning models, e.g. via curriculum learning [Elman, 1993, Bengio et al., 2009], has historically yielded mixed results. We speculate that one of the reasons is that approaches largely relied on ranking by difficulty examples within a dataset [Wu et al., 2020, Lalor and Yu, 2020]. Not only is accurately estimating difficulty challenging, but datasets may also lack the diversity or granularity needed to isolate and teach the fundamental "skills" required for complex problem-solving. Recent studies suggest that curriculum learning is more effective when "decomposed" datasets that isolate atomic skills are available to the learner [Lee et al., 2024].

We build upon this intuition and, assuming access to a teacher model that can reason step-by-step, we propose a way of "decomposing" a dataset of mathematical problems into a hierarchy of problems with different difficulty levels. Our approach, DECOMPX, enables smaller LMs to progressively acquire elemental skills before tackling more intricate ones. Although we focus on mathematical datasets such as MATH [Hendrycks et al., 2021] and AIME [of America, 2024], as well as code generation datasets such as [Chen et al., 2021], we think the approach might be applied to a broader range of domains in the future.

39th Conference on Neural Information Processing Systems (NeurIPS 2025).

Our dataset decomposition approach starts with a dataset of mathematical questions and step-by-step solutions. The main idea is that a teacher model can generate simpler problems by looking at sub-steps in the solution. Every sub-step is by definition simpler than the original problem and thus can be used as the answer for a latent, simpler problem. Crucially, this process can be recursively iterated: we can now ask a teacher model to create a step-by-step solution to the sub-problem and recursively generate simpler problems in the same manner until a stopping criterion – such as recursion depth or problem simplicity – is met. Given the strong inductive bias of step-by-step reasoning, the teacher is encouraged to work out solutions to progressively simpler problems. This process creates a tree of sub-problems for each question in the original dataset.

To facilitate curriculum construction, we assign difficulty scores to sub-problems based on their structural complexity. We tag sub-problems and construct a graph of tags by connecting each sub-problem's tag to its parent problem's tag. The graph synthesizes tag relationships across the entire dataset. The simplicity of a problem is then computed both using the "depth" of the tags associated with a given problem and the number of sub-problems it generates.

We use our dataset decomposition to train small LMs to do mathematical reasoning and code generation via supervised fine-tuning (SFT). Even when decomposed problems are presented i.i.d. to the models, our approach improves the ability of small LMs to learn from a small set of examples. We see further gains when sub-problems are presented in the order of increasing complexity.

Apart from the dataset augmentation and curriculum construction aspect, one of the useful byproducts of our method is the potential for creating a "cartography" of a given dataset [Swayamdipta et al., 2020]. Our induced graph of skills can be used to gain more insights into coverage of the dataset and how additional samples might increase the coverage, or even how two seemingly different datasets are related to each other.

## 2  Related Work

**Data Augmentation**  Data Augmentation, synthetically generated from an LLM can be seen as a form of distillation [Hinton et al., 2015, Kim and Rush, 2016]. This approach has been especially successful in distilling reasoning capabilities into smaller models [Mitra et al., 2023, Li et al., 2023, Mitra et al., 2024]. This approach has also been applied to generate mathematical reasoning traces. For instance, Self-Taught Reasoner [Zelikman et al., 2022] generates and then learns from its own chain-of-thought reasoning steps. MuggleMath [Li et al., 2024] and MetaMath [Yu et al., 2024b] both amplify diversity by evolving problem queries and sampling multiple reasoning traces, and by applying paraphrases and backward reasoning transformation, respectively. MuMath [Yin et al., 2024] further enriches examples through multiple rewrites using several "perspectives", e.g., paraphrase, symbolic reformulation. MathGenie [Lu et al., 2024] back-translates a small seed set through a generator–verifier loop to create high-fidelity new problems. PersonaMathQA [Luo et al., 2024] adds "persona diversification" plus self-reflection to generate richer problem contexts. Beyond direct data augmentation, Shridhar et al. [2023] generates subquestions and solutions to distill the knowledge from the teacher model to students, Huang et al. [2025] synthesizes QA pairs by extracting key points from problems. To simultaneously address the quality, diversity, and complexity of the dataset, Davidson et al. [2025] proposes Simula, a unified framework for generating and evaluating synthetic data. However, existing data augmentation techniques somewhat neglect prior knowledge and difficulty. To address this gap, we propose hierarchically decomposing problems with multi-step augmentation so models progressively acquire elemental skills.

**Curriculum Learning**  Bengio et al. [2009], Elman [1993] originally propose to apply curriculum to train LMs. Wu et al. [2020] shows mixed results when applying curriculum learning methods based on example difficulty. In the past, many have studied different curriculum learning strategies to either increase the context length or more efficiently train LMs [Press et al., 2020, Nagatsuka et al., 2021], with unconvincing results. More recently, studies started to investigate synthetic data creation associated with curriculum learning, yielding promising results for training small LMs for code [Naïr et al., 2024]. Work on TinyStories [Eldan and Li, 2023] shows that carefully synthetically curated datasets can be helpful in teaching tiny LMs basic English proficiencies. These works align with ours in the hypothesis that the synthetic data creation of easier examples can be used as a driving factor underpinning a successful curriculum. In the robotics domain, [Florensa et al., 2017] proposes a "reverse" learning strategy that starts RL training from the goal state and gradually guides the policy

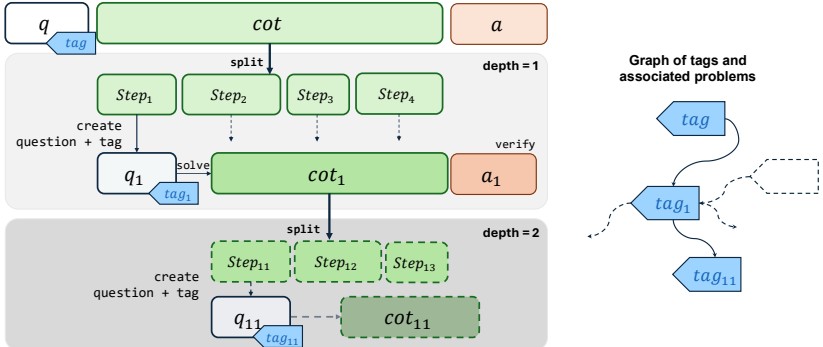

*Figure 1. Left:* We recursively decompose a math example $(q, cot, a)$ into a set of smaller problems (depth 2 in the figure). We first split the *cot* into steps, then create a question for each step and an associated concept tag. We then ask the teacher model to solve the question step-by-step. We verify the final answer by ensuring it is the same as the answer obtained without the ground-truth step in context. We then recursively apply this procedure until a stopping criterion. *Right:* We create a graph of tags, where dependency relation is given by the hierarchy in the decomposition tree. The graph of tags is used to quantify the difficulty of a generated sub-problem.

to learn to reach the goal from a set of start states increasingly far from the goal. This approach also builds a curriculum with increasing difficulty levels (in a guaranteed way) by reversing the original task. However, our reasoning domain is different because it allows us to leverage the compositionality of language to decompose the reasoning tasks at much meaningful abstraction levels.

# 3 DECOMPX: Dataset Decomposition

In this section, we propose a recursive algorithm, DECOMPX, that decomposes complex math problems into simpler subproblems based on their underlying reasoning steps. Our algorithm consists in two phases: first, we decompose each example into a hierarchical structure of sub-problems; second, we connect these sub-problems at a dataset level using a tagging approach and we build a graph of tags, useful to infer the difficulty of every sub-problem. Finally, we describe how we use the difficulty score in our curriculum learning procedure.

## 3.1 How to generate subproblems? - Recursive Problem Decomposition

We propose a recursive dataset decomposition framework that constructs verified, grounded subproblems from multi-step solutions. Each sub-problem corresponds to a clear, atomic mathematical reasoning operation, explicitly linked to a core mathematical concept and validated for correctness. We assume access to a teacher model $\mathcal{T}$ (a large LM, we use GPT-4o in our experiments), which we assume is proficient at the task of the dataset of interest.

**Step Extraction.** Given a solution trace *cot* for a math problem $q$, we first decompose it into at most $k$ reasoning steps. This is achieved by prompting the teacher model to segment the text based on conceptual granularity: $cot \mapsto [s_1, s_2, \dots, s_k]$, where each step $s_i$ introduces a new and distinct mathematical operation.

**Concept Tagging.** For each step $s_i$, we extract an atomic concept tag $t_i$ by querying the teacher model to identify the most specific mathematical concept that governs the reasoning step.

**Subproblem Generation.** Given the original problem $q$, a step $s_i$, and its tag $t_i$, we generate a new subproblem $q_i$ (i.e. a question) grounded in the original context $(t_i, q)$. Then, we ask the teacher model to solve the generated problem $q_i$ step-by-step leading to a solution $cot_i$ and final answer $a_i$.

**Verification.** To assess that the answer to the generated sub-problem $q_i$ is correct and answerable, we also ask the teacher model to solve $q_i$ without access to the original context $(t_i, q)$ resulting in an answer $\hat{a}_i$. Given that $a_i$ has been generated with access to the original context, it is more likely to be correct. Therefore, we compare the resulting numerical answer $\hat{a}_i$ to $a_i$ using a symbolic verifier $\mathcal{V}$. Only subproblems satisfying $\mathcal{V}(a_i, \hat{a}_i) = \text{True}$ are accepted. Otherwise, the subproblem is regenerated with a retry budget of $R$ attempts.

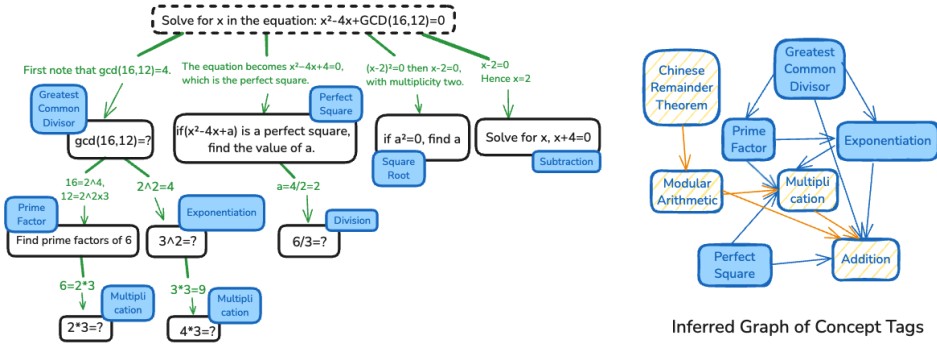

Decomposition of an example

Inferred Graph of Concept Tags

*Figure 2.* Left: We show the decomposition of the math problem on the top obtained with our method, along with the associated concept tags. The problems in the solid white boxes are the generated sub-problems. Right: The graph of concept tags, obtained by connecting the tags across the dataset examples.

**Recursive Expansion.** The sub-problem reasoning $cot_i$ contains further multi-step reasoning by construction. Therefore, we can apply the process recursively by decomposing $cot_i$ using the same procedure up to a maximum depth $D$. This produces, for every example in the dataset, a nested structure:

$$q_i, cot_i, a_i \mapsto \{(q_{i,j}, cot_{i,j}, a_{i,j})\}_{j=1}^{m},$$

where each child $q_{i,j}$ corresponds to a problem grounded in the constituent sub-step $j$ of $cot_i$.

In summary, the final output is a dataset composed of sub-problems, associated concept tags, sub-problem depth, reasoning steps and answers (see Figure 1, left). We also illustrate the whole workflow with concrete problems in Figure 2.

---

**Algorithm 1** Recursive Dataset Decomposition

---

**Input** : Problem set $\mathcal{D}_{\text{raw}}$, teacher model $\mathcal{T}$, maximum decomposition depth $D$, maximum steps per layer $k$
**Output** : Decomposed dataset $\mathcal{D}_{\text{curr}}$

1  Initialize empty dataset $\mathcal{D}_{\text{curr}} = \{\}$
2  **Function** DecomposeExample($q, cot, a, d, \mathcal{D}_{\text{curr}}$)
3      Split $cot$ into steps $\{s_1, \dots, s_k\}$ using $\mathcal{T}$
4      **for** $i = 1$ **to** $k$ **do**
5          Generate concept tag $t_i$ for step $s_i$ using $\mathcal{T}$
6          **for** $retry = 1$ **to** *MaxRetries* **do**
7              Generate question $q_i$ from $(s_i, t_i)$
8              Generate step-by-step solution $cot_i$ and answer $a_i$ from $(q, s_i, t_i)$ using $\mathcal{T}$
9              Verify $a_i$ via auto-solver and consistency check *(see paper)*
10             **if** *verified* **then**
11                 Break
12             **end**
13         **end**
14         $\mathcal{D}_{\text{curr}} = \mathcal{D}_{\text{curr}} \cup (q_i, cot_i, a_i, t_i, d)$
15         **if** $d < D$ **then**
16             DecomposeExample($q_i, cot_i, a_i, d+1, \mathcal{D}_{\text{curr}}$)  *# Recursive call*
17         **end**
18     **end**
19 **end**
20 **foreach** $(q, cot, a) \in \mathcal{D}_{\text{raw}}$ **do**
21     DecomposeExample($q, cot, a, 0, \mathcal{D}_{\text{curr}}$)
22     $\mathcal{D}_{\text{curr}} = \mathcal{D}_{\text{curr}} \cup (q, cot, a, -, D)$
23 **end**

---

## 3.2 How difficult is a certain (sub)problem? – Concept Dependency Graph

We construct a directed acyclic graph (DAG) over concepts derived from the recursive decomposed problem-solving process above. This graph encodes the prerequisite relationships between mathematical operations across all the examples in the dataset, thus enabling curriculum design.

Let $\mathcal{G} = (\mathcal{V}, \mathcal{E})$ denote the *Concept Dependency Graph*, where $\mathcal{V}$ is a set of concept tags (such as "GCD" or "Square Root"), and $\mathcal{E} \subseteq \mathcal{V} \times \mathcal{V}$ is a set of directed edges representing prerequisite relationships between concepts. A directed edge $(u, v) \in \mathcal{E}$ indicates that concept $v$ depends on concept $u$.

The construction algorithm proceeds as follows. We initialize an empty directed graph $\mathcal{G}$. We add the ensemble of tags across examples to the node set $\mathcal{V}$. A parent tag $t_p$ has a child tag $t_c$ if they are obtained while decomposing the same example and their depth differ by 1. For every parent tag $t_p$ and child tag $t_c$, we insert an edge $(t_p, t_c) \in \mathcal{E}$, unless $t_p = t_c$.

Nodes with zero in-degree in $\mathcal{G}$ represent foundational concepts. Formally, the set of root nodes is defined as $\mathcal{R} = \{v \in \mathcal{V} \mid \text{in-degree}(v) = 0\}$. To quantify concept difficulty, we define a depth function $d : \mathcal{V} \to \mathbb{N}$ such that:

$$d(v) = \begin{cases} 0 & \text{if } v \in \mathcal{R}, \\ 1 + \max_{(u,v) \in \mathcal{E}} d(u) & \text{otherwise.} \end{cases}$$

This depth serves as a proxy for the reasoning complexity required to apply concept $v$.

### 3.2.1 Concept Clustering via Embedding Similarity

Our generated tags include variations such as "GCD" and "Greatest Common Divisor" that refer to the same mathematical concept. To unify these concepts and minimize semantic redundancy in our graph, we use unsupervised clustering on tags' embedding representations.

Specifically, each concept tag $t \in \mathcal{V}$ is mapped to a dense vector representation $\phi(t) \in \mathbb{R}^d$ using a pre-trained LM. We then compute pairwise cosine similarities between all tag embeddings. Denote, the similarity as $S(i, j)$.

To cluster tags, we employ a greedy clustering algorithm with a predefined similarity threshold $\delta$. Specifically, we sequentially select unassigned tags as cluster representatives and group all other tags exceeding the similarity threshold under them. Formally, the mapping from original tags to cluster representatives $\tau$ is concisely defined as:

$$\tau(t_j) = \arg \max_{t_i \in \mathcal{V}_{\text{rep}}} \{S(t_j, t_i) \mid S(t_j, t_i) \geq \delta\},$$

where $\mathcal{V}_{\text{rep}} \subseteq \mathcal{V}$ represents the set of selected representative tags.

After clustering, we relabel the nodes in the concept dependency graph $\mathcal{G}$ accordingly. Edges are rewired based on these updated labels, explicitly removing any self-loops arising from clustering:

$$\mathcal{E}' = \{(\tau(u), \tau(v)) \mid (u, v) \in \mathcal{E}, \tau(u) \neq \tau(v)\}.$$

This results in a refined concept dependency graph $\mathcal{G}' = (\mathcal{V}', \mathcal{E}')$, where each node uniquely represents a distinct conceptual skill.

### 3.2.2 Difficulty Measurement via Structural and Conceptual Features

The difficulty of a data sample stems from both the number of mathematical operations it includes and the complexity of the concepts involved. For example, consider solving the following three problems: a) 1+1; b) 1+1+...+1; (sum of thousands of 1); c) `sqrt(3) + (5/76)**2`. c) is harder than P1 due to conceptual complexity (the complexity of operations involved) while b) is harder than a) because of structural complexity, the number of atomic operations involved.

While the Concept Dependency Graph provides a data-driven way to gauge a concept's complexity by measuring the depth of the concept's tag in the graph, it may fall short in accounting for the structural

complexity of a specific reasoning step that involves that concept, such as the number of sub-problems it can be decomposed into, or how important the sub-problem is to solve other problems in the dataset. To address this, we introduce a composite difficulty score that combines both conceptual and structural factors for a more comprehensive characterization.

Given a reasoning step $s$ (or a data sample $q$ before the first iteration of decomposition) in the recursive decomposition tree (e.g., $step_1$ in Figure 1), we compute:

- **structural complexity** $SC(s)$: the number of direct children of the reasoning step, reflecting its structural branching factor. In the example in Figure 1, $SC(step_1) = 3$ because $step_1$ can be further decomposed into three lower-level reasoning steps (i.e., $step_{11}$, $step_{12}$, and $step_{13}$).
- **conceptual depth** $CD(s)$: the maximum depth of the concept tag in the Concept Dependency Graph $\mathcal{G}$ that is associated with this reasoning step. In the example in Figure 1, $CD(step_1)$ equals the depth of $tag_1$ in $\mathcal{G}$.

Therefore, the overall **difficulty score** $\ell(s) \in \mathbb{R}^+$ is defined as a weighted combination of these two terms:

$$\ell(s) = \alpha_1 \cdot SC(s) + \alpha_2 \cdot CD(s),$$

where $\alpha_1, \alpha_2 \in \mathbb{R}^+$ are tunable coefficients that balance structural complexity and conceptual depth.

### 3.3 Curriculum Learning via Difficulty Measurement

To leverage the difficulty scores $\ell$ during training, we implement a staged curriculum learning framework where the model is exposed to data in increasing order of difficulty. This approach enables the model to first acquire capabilities on simpler sub-problems before attempting harder examples.

Let $\mathcal{D}' = \{q, cot, \ell(q)\}$ be the decomposed dataset where the question in each sample is annotated with our difficulty score. We partition $\mathcal{D}'$ into $K$ non-overlapping subsets $\mathcal{D}'_1, \ldots, \mathcal{D}'_K$ based on the quantiles of difficulty:

$$\mathcal{D}'_1 \cup \cdots \cup \mathcal{D}'_K = \mathcal{D}',$$
$$\mathcal{D}'_j = \{(q, cot) \in \mathcal{D}' \mid qb_{j-1} \leq \ell(q) < qb_j\},$$

where $\{qb_0, qb_1, \ldots, qb_K\}$ are the quantile breakpoints computed from the score distribution. We adopt an *easy-to-hard* curriculum, where the model is trained sequentially from $\mathcal{D}'_1$ to $\mathcal{D}'_K$. Each stage is trained with early stopping to avoid overfitting and to allow controlled progression:

$$\text{Stage } i : \quad \theta_i \leftarrow \arg \min_\theta \mathcal{L}(\theta; \mathcal{D}'_i),$$

where $\theta_i$ denotes model parameters after stage $i$. We split the training budget across the full curriculum, so that experiments presented in this paper are compute matched for a given setting.

## 4 Experiments and Results

In this section, we describe our evaluation procedure used to validate the effectiveness of DECOMPX.

### 4.1 Experimental Setup

**Models** We adopt the Qwen2.5-1.5B [Qwen et al., 2024] and Qwen3-4B-Base [Qwen et al., 2025] models as our student models. Our dataset decomposition is driven by GPT-4o and o4-mini for mathematical reasoning and code generation, respectively, which we leverage as our teacher models.

**Datasets** Our setup uses MATH [Hendrycks et al., 2021] and the American Invitational Mathematics Examination (AIME). MATH [Hendrycks et al., 2021] is a benchmark of competition math problems of varying difficulty. We evaluate on the same 500 samples in the prior work [Lightman et al., 2023]. AIME contains challenging mathematical competition problems. For training, we use AIME '24 as training set and AIME '25 as test set. Both datasets contain 30 problems that were used in the AIME in 2024 and 2025, respectively. More details can be found in Appendix A.

*Table 1.* Comparison of testing accuracy to LLMs on the MATH-500 benchmark.# data refers to the number of examples used for fine-tuning.    ‡We evaluate based on the checkpoint released at https://huggingface.co/Qwen/Qwen2.5-1.5B using lighteval [Habib et al., 2023].

| Model | # data | MATH-500 |
|---|---|---|
| *Open-Weights Models* | | |
| Qwen2.5-1.5B [Qwen et al., 2024] | N.A. | 35.0 |
| Qwen2.5-3B [Qwen et al., 2024] | N.A. | 42.6 |
| Llama-3-70B [Dubey et al., 2024] | N.A. | 42.5 |
| Mixtral-8x22B [Jiang et al., 2024] | N.A. | 41.7 |
| Gemma2-27B [Team et al., 2024] | N.A. | 42.7 |
| MiniCPM3-4B [Hu et al., 2024] | N.A. | 46.6 |
| Gemma2-9B-Instruct [Team et al., 2024] | N.A. | 44.3 |
| Llama3.1-8B-Instruct [Dubey et al., 2024] | N.A. | 51.9 |
| *Qwen2.5-1.5B SFT on MATH* | | |
| Base‡ | N.A. | $47.2 \pm 2.2$ |
| SFT (full dataset) | 7500 | $47.6 \pm 2.2$ |
| SFT | 360 | $48.4 \pm 2.2$ |
| SFT-Direct Distillation | 360 | $49.6 \pm 2.2$ |
| SFT-MetaMATH-Aug | 2638 | $37.2 \pm 6.8$ |
| SFT-MuggleMath | 147787 | $50.4 \pm 2.2$ |
| SFT-DecompX (Ours) | 4500 | $50.8 \pm 2.2$ |
| SFT-DecompX + Curriculum (Ours) | 4500 | $51.6 \pm 2.2$ |

To further demonstrate the generalization capability and versatility of our method, we conduct additional experiments in a non-mathematical domain that demands complex reasoning: coding. Specifically, we use the CodeForces-CoTs dataset [Penedo et al., 2025] (competitive programming solutions in C++) from HuggingFace Open-R1 [Hugging Face, 2025] for training. This dataset contains solutions generated by DeepSeek-R1 [DeepSeek-AI et al., 2024], where long and complex reasoning traces are common. For evaluation, we employ the HumanEval benchmark [Chen et al., 2021] (Python function completion), which serves as an explicitly out-of-distribution scenario.

**Training Details**    Follow the fine-tuning setup in the previous work [Muennighoff et al., 2025] we train each model for 5 epochs with a batch size of 16. We train the models using `bfloat16` precision with a learning rate of $10^{-5}$, warmed up linearly for 5% and then decayed to 0 over the rest of the training, following a cosine schedule. We use the AdamW optimizer [Loshchilov and Hutter, 2019]. Unless otherwise specified, we evaluate with a temperature of 0 (greedy decoding) and measure accuracy (equivalent to pass@1). The results are averaged over three different training seeds. Our experiments are conducted on NVIDIA A100 GPUs with 80GB VRAM.

**Baselines**    We compare our dataset decomposition and curriculum learning method with a set of baseline systems: (1) closed-weights models such as GPT-4o; (2) open-weights models; (3) Supervised Fine-Tuning (SFT), which uses the training set of the original datasets; (4) direct distillation from the same teacher model and the same original datasets. (4) Data augmentation methods including MetaMath [Yu et al., 2024a] and MuggleMath [Li et al., 2024]: We applied MetaMath's augmentation pipeline for mathematical datasets by rephrasing questions as well as generating answers in four augmentation types (rephrasing, self-verification, answer-augmentation and backward reasoning).

## 4.2  Main Results

We present our main results, evaluated across different benchmarks, and compared with baseline approaches. We summarize the findings below.

*Table 2.* Comparison of testing accuracy to LLMs on the AIME 2025 benchmark. # data refers to the number of examples used for fine-tuning. ‡We evaluate based on the checkpoint released at https://huggingface.co/simplescaling/s1-32B without budget forcing.

| Model | # data | AIME2025 |
|---|---|---|
| ***Open-Weights Models*** | | |
| Qwen2.5-72B-Instruct [Qwen et al., 2024] | N.A. | 15.0 |
| S1-32B‡ [Muennighoff et al., 2025] | 1000 | 13.3 |
| ***Close-Source Models*** | | |
| GPT-4o [OpenAI et al., 2023] | N.A. | 7.6 |
| ***Qwen3-4B-Base SFT on AIME2024*** | | |
| Base | N.A. | $10.0 \pm 5.6$ |
| SFT | 30 | $3.3 \pm 3.3$ |
| SFT-Direct Distillation | 30 | $6.7 \pm 4.6$ |
| SFT-MetaMATH-Aug | 114 | $4.4 \pm 4.2$ |
| SFT-DecompX (Ours) | 385 | $13.3 \pm 6.3$ |
| SFT-DecompX + Curriculum (Ours) | 385 | $16.7 \pm 6.9$ |
| ***DeepSeek-R1-Distill-Qwen-1.5B SFT on MATH*** | | |
| Base | N.A. | $23.33 \pm 7.85$ |
| SFT-DecompX (Ours) | 4500 | $30.00 \pm 8.51$ |

**DECOMPX improves performance across different benchmarks.** We start our analysis by investigating performance on smaller LLMs. We see that across two different base models and datasets, DECOMPX shows consistent gains over standard baselines; Table 1 presents results using Qwen2.5-1.5B finetuned and evaluated on MATH. We note that it achieves a 2.4% improvement over SFT and a 13.6% relative gain over SFT-MetaMATH-Aug. It also demonstrates the advantages of DE-COMPX over direct distillation. Table 2 reports test accuracy on AIME2025 obtained from fine-tuning Qwen3-4B-Base on the AIME2024 data. Again, DECOMPX performs well, showing improvements of 10% over SFT and 8.9% over SFT-MetaMATH-Aug. It even outperforms Qwen2.5-72B-Instruct, a significantly larger model, using only 385 training samples. Overall, these results highlight the effectiveness of our method and validate the benefit of structured decomposition and curriculum learning in mathematical reasoning tasks. Finally, we note that DECOMPX is better than MetaMATH on both of the benchmarks, which is used for generating augmented data for finetuning. MuggleMath also underperforms our approach despite leveraging substantially more training data and richer prior knowledge from the seed datasets (MATH and GSM8K [Cobbe et al., 2021]). These results further support the advantage of our structured, decomposition-based curriculum. Last but not least, Table 3 demonstrates the advantage of our methods compared to the baselines on the code generation tasks.

**Performance reduced during post-training for traditional SFT.** Surprisingly, we find that SFT on mathematical datasets may reduce the performance compared to the base model in our experiments. This behavior is especially visible on AIME (Table 2), suggesting that the model may be prone to overfitting given the small dataset size.

**DECOMPX shows better generalisation.** S1 [Muennighoff et al., 2025] is a reasoning model obtained via supervised finetuning based on Qwen2.5-32B-Instruct using 1,000 samples. Although this sample-efficient baseline achieves strong performance on MATH-500 and AIME2024, it does not perform as well on AIME2025. The model is over-parameterized relative to the amount of signal in the data. which means that the dataset contains less information than the model can represent. In contrast, DECOMPX outperforms S1 even with a much smaller base model (4B) on AIME2025. Model trained only on decomposed data generated from AIME2024 performance suggests its effectiveness in generalizing to unseen mathematical tasks. SFT on AIME 2024 fails to generalize to another year of AIME, whereas our data decomposition can help model better to learn the features that stay predictive not only for in-domain generalisation, but also under the distribution shift. With curriculum learning added to DECOMPX, it can reshape the training trajectory so that the model first captures

*Table 3.* Comparison of testing accuracy to LLMs on the HumanEval benchmark. # data refers to the number of examples used for fine-tuning.

| Model | HumanEval pass@1 |
|---|---|
| ***Open-Weights Models*** | |
| Llama3-8B [Dubey et al., 2024] | 33.5 |
| Mistral-7B [Jiang et al., 2024] | 29.3 |
| Gemma2-9B [Team et al., 2024] | 37.8 |
| ***Qwen2.5-1.5B-Instruct SFT on Codeforces-CoTs*** | |
| Base | $34.15 \pm 3.71$ |
| SFT | $35.37 \pm 3.74$ |
| SFT-DecompX (Ours) | $42.68 \pm 3.87$ |
| ***DeepSeek-R1-Distill-Qwen-1.5B SFT on Codeforces-CoTs*** | |
| Base | $36.01 \pm 1.85$ |
| SFT-DecompX (Ours) | $57.90 \pm 0.98$ |

universal mathematical skills, then gradually adapts itself against wording and distribution drift. We see evidence that curriculum learning can lead to better generalization and help model competitive performance on AIME 2025 even when the other strong baselines do not.

**Curriculum learning is beneficial.** In regular SFT, the examples are randomly shuffled. However, with the sample difficulty measurement yielded by DECOMPX, we can create a learning curriculum, starting with the easiest samples and progressively moving towards harder ones. This explores whether difficulty measurements are useful in forming a curriculum without changing the set of training examples. From Table 1 and Table 2, we find that even when training on the same set of examples, difficulty measurements are useful for improving performance, compared to training with a random sample order. Indeed, curriculum yields a 0.8% and 3.4% on MATH and AIME datasets for DECOMPX, or relative improvements of 1.6% and 25.6% respectively over randomly ordered sampling.

### 4.3 Case Study

Table 4 presents both the original data sample from the MATH training set (Left) and three corresponding generated data samples (Right), which are decomposed based on the original example. Using the difficulty measurement defined earlier, we compute the difficulty scores and categorize the samples into different levels. In both the MATH and AIME datasets, the difficulty scores range from 2 to 20. We define three levels of difficulty: low (scores around 2 to 6), medium (around 6 to 10), and high (from 10 up to 20). As shown in Table 4, we successfully decompose complex problems into simpler subproblems and effectively quantify the difficulty of each subproblem.

## 5 Discussion and Conclusion

In this work, we propose a novel curriculum learning approach via recursive dataset decomposition, enabling smaller language models to progressively master mathematical reasoning tasks. Our experiments on math benchmarks (MATH and AIME) show significant performance improvements over baseline methods, highlighting the effectiveness of our structured decomposition and difficulty-scoring strategies. In the future, we plan to improve DECOMPX so it could be effectively used for generating more general datasets in a broader range of domains. However, this will require extensive study on teacher models' capability and reliability in task decomposition in other specific domains.

**Implications and future work.** Our work provides stronger evidence for self-improvement. While our current work adopts a "large teacher to small student" setup, the method was designed to leverage the previous generation of models not only as data consumers but also as data generators. The key idea is to create a high-quality synthetic curriculum that teaches complex concepts by shaping data for learning, rather than simply augmenting the dataset.

*Table 4.* Examples of an original data sample from MATH training set and its generated data samples.

| Original Data | Decomposed Data |
|---|---|
| **Problem:** Simplify $\frac{3^4+3^2}{3^3-3}$. Express your answer as a common fraction. **Solution:** The common factor of 3 in the numerator and the denominator can be factored out: $\frac{3^4+3^2}{3^3-3} = \frac{3(3^3+3)}{3(3^2-1)} = \frac{3^3+3}{3^2-1}$. Now compute: numerator is $27 + 3 = 30$, denominator is $9 - 1 = 8$, so $\frac{30}{8} = \boxed{\frac{15}{4}}$. | **Generated Data with Lower Difficulty**   **Difficulty Score: 4.0** **Problem:** What is the value of $3^3$? **Solution:** To solve for $3^3$, we multiply 3 by itself three times: $3 \times 3 = 9$, then $9 \times 3 = 27$. So, $3^3 = 27$. **Tag:** Exponentiation **Generated Data of Medium Difficulty**   **Difficulty Score: 10.0** **Problem:** What is the greatest common divisor (GCD) of the numbers 30 and 8? **Solution:** To find the greatest common divisor (GCD) of 30 and 8, we need to identify the largest number that divides both 30 and 8 without leaving a remainder. 1. List the factors of 30:- 1, 2, 3, 5, 6, 10, 15, 30  2. List the factors of 8:- 1, 2, 4, 8  3. Identify the common factors of 30 and 8:- The common factors are 1 and 2.  4. The greatest of these common factors is 2. **Tag:** GCD Calculation **Generated Data with Higher Difficulty**   **Difficulty Score: 18.0** **Problem:** Simplify the expression $\frac{3^3+3}{3^2-1}$. **Solution:** We first compute the powers $3^3 = 27, 3^2 = 9$. Next, substitute into the expression: $\frac{27+3}{9-1}$. Perform addition and subtraction $\frac{30}{8}$. And simplify the fraction $\frac{30}{8} = \frac{15}{4}$. Final Answer is $\boxed{\frac{15}{4}}$. **Tag:** Fraction Simplification |

This design naturally supports scenarios where teacher and student have similar capacity: the benefit comes not from the size gap but from the teacher's ability to organize and distill knowledge into teaching-oriented examples. In the weak-to-strong generalization setting, this can be achieved as long as the weaker model is already capable of producing clean and well-structured decompositions. We leave this as a promising future direction, especially at scale, where our proposed hierarchical data decomposition and curriculum learning could have even greater impact.

**Limitations and broader impacts.** This work proposes using stronger LLM teachers to recursively generate simpler data that builds up a curriculum for training student models. It assumes access to strong enough teacher models that are capable of understanding a math problem, decomposing the problem into meaningful sub-tasks, and faithfully describing the sub-tasks. As mentioned above, for tasks beyond math, where the reasoning path to solve a task is less divisible in an obvious way, the teacher models may face challenges in generating the curriculum. This may require the design of better scaffolding and/or the use of more advanced teacher models. Moreover, LLMs have been shown to hallucinate in various ways, our method is inherently vulnerable because LLM usage is at the core of the system design. For example, a hallucinating or fabricating teacher model may generate inaccurate reasoning chains and decompose them in the wrong ways. The student models trained on the generated curriculum in such a way may result in poor performance or learn unexpected behaviors due to the suboptimality of the curriculum. Finally, we acknowledge that our work is not yet at a stage to be used in many real-world tasks, especially in domains involving high-risk decision making such as law enforcement, legal, finance, or healthcare.

## Acknowledgement

The authors are deeply grateful to Eric Yuan and Marc-Alexandre Côté for their invaluable help. We would also like to thank Colin Raffel, Matthew Macfarlane, Ayush Agrawal, Gyung Hyun Je, Vedant Shah and Zhihao Zhan for many stimulating and helpful discussions.

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

# Contents

# A    Details of subset of MATH training data

In Table 5, we present the original and sampled sizes of the MATH dataset used in our experiments, broken down by subject domain and difficulty level. Sampling was performed uniformly at random within each group to ensure representative coverage of the various topics and levels.

*Table 5.* **Statistics of the original and sampled MATH dataset by subject domain (left) and by difficulty level (right).** Samples were drawn uniformly at random within each group to ensure representative coverage across topics and difficulty tiers.

| By Domain | | | | By Difficulty Level | | |
|---|---|---|---|---|---|---|
| Domain | #Total | #Sampled | | Level | #Total | #Sampled |
| Algebra | 1744 | 84 | | Level 1 | 566 | 26 |
| Counting and Probability | 771 | 36 | | Level 2 | 1348 | 65 |
| Geometry | 870 | 43 | | Level 3 | 1592 | 77 |
| Intermediate Algebra | 1295 | 63 | | Level 4 | 1690 | 81 |
| Number Theory | 869 | 41 | | Level 5 | 2304 | 111 |
| Prealgebra | 1205 | 58 | | | | |
| Precalculus | 746 | 35 | | | | |
| ***#Sampled / #Total*** | | | | ***300 / 7500*** | | |

# B    Examples of Concept Dependency Graph

Figure 3 presents the concept dependency graphs constructed during the data decomposition process. We observe that an atomic mathematical operation, such as *Addition*, has many edges linking it to more advanced operations.

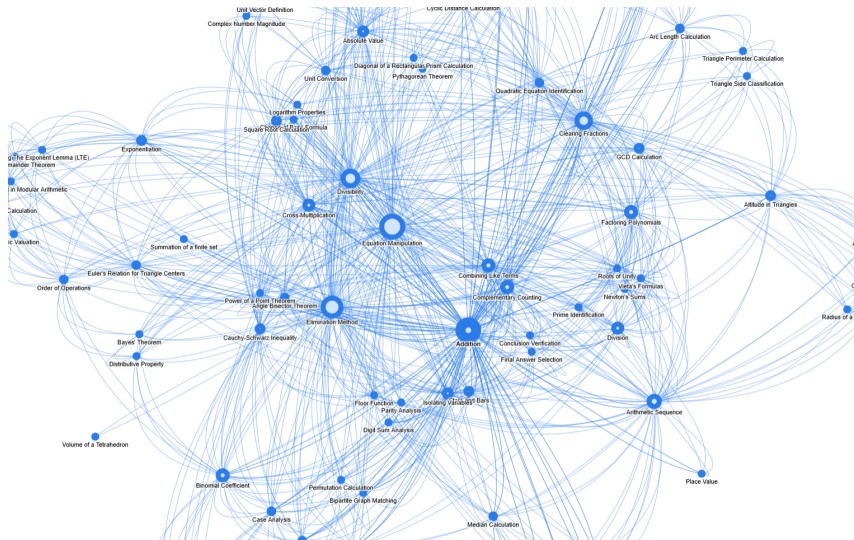

*Figure 3.* **Concept dependency graph constructed during the AIME data decomposition process.** Nodes represent mathematical concepts, and edges indicate prerequisite relationships between concepts.

# C    Zero-shot Small Model Performance

We partitioned the decomposed AIME2024 dataset into five equal-sized bins (quintiles) based on our proposed difficulty measurement, shown in Table 6. This measurement is derived from the concept dependency graph, designed to reflect the conceptual complexity of each problem. We then evaluated the zero-shot performance of the Qwen3-4B-Base model on each difficulty tier.

Our results shown in Figure 4 demonstrate a clear inverse correlation between difficulty score and model accuracy: the model achieves the highest accuracy on problems with the lowest difficulty scores (Quintile 1), and its performance degrades as the difficulty increases, reaching the lowest accuracy on the highest difficulty tier (Quintile 5). This performance trend validates the effectiveness of our concept dependency graph-based difficulty metric in capturing the relative hardness of mathematical problems.

*Table 6.* **Definition of difficulty quintiles based on concept dependency graph scores.** Each quintile groups problems whose scores fall within the specified range.

| Quintile | Difficulty Score Range |
|---|---|
| Quintile 1 (Easiest) | 2.0 – 4.0 |
| Quintile 2 | 4.0 – 4.0 |
| Quintile 3 | 4.0 – 6.0 |
| Quintile 4 | 6.0 – 10.0 |
| Quintile 5 (Hardest) | 10.0 – 20.0 |

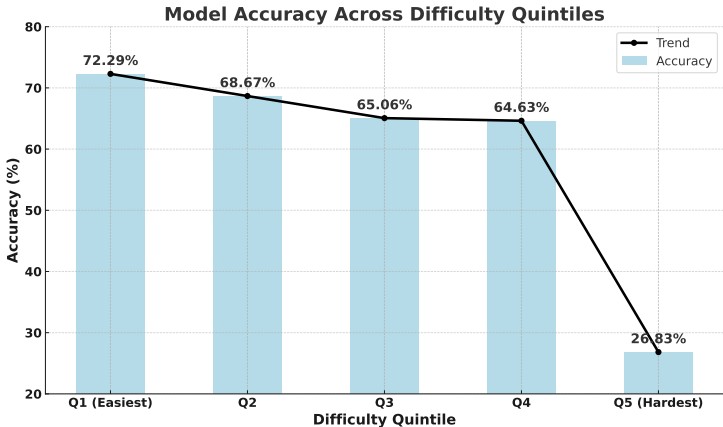

*Figure 4.* **Zero-shot performance of the Qwen3-4B-Base model across difficulty quintiles.** Accuracy decreases as problem difficulty increases, validating our proposed difficulty metric.

## D  Tag Clustering Details

We summarize the tag information identified in both the MATH dataset (Table 7) and the AIME dataset (Tables 8 and 9).

## E  Examples of Decomposed Data

We apply our data decomposition method to the S1 dataset and demonstrate the two-layer recursive decomposition process for a sample data point in Table 10.

*Table 10.* **Decomposed S1 data sample.** Black text denotes the original data; yellow text represents the first level of decomposition; and blue text represents the second level of decomposition.

{Question:"Given a rational number, write it as a fraction in lowest terms and calculate the product of the resulting numerator and denominator. For how many rational numbers between 0 and 1 will 20! be the resulting product?",

Solution:"Let the rational number between 0 and 1 be $r$. When written as a fraction in lowest terms, let it be $\frac{a}{b}$, where $a$ and $b$ are positive integers, $\gcd(a, b) = 1$, and $0 < a < b$.We are given that the product of the numerator and denominator is 20!, so $a \times b = 20!$.We need to find the number of pairs $(a, b)$ such that $a \times b = 20!$, $\gcd(a, b) = 1$, and $a < b$.Let $N = 20!$. The prime factorization of $N$ is $p_1^{e_1} p_2^{e_2} \cdots p_k^{e_k}$, where $p_i$ are distinct prime numbers.The number of distinct prime factors of 20! is the number of primes less than or equal to 20. These primes are 2, 3, 5, 7, 11, 13, 17, 19. Thus, there are 8 distinct prime factors, so $k = 8$.Since $a \times b = N$ and $\gcd(a, b) = 1$, each prime power factor $p_i^{e_i}$ of $N$ must be entirely a factor of $a$ or entirely a factor of $b$.The number of ways to distribute the prime power factors between $a$ and $b$ is $2^k$, where $k$ is the number of distinct prime factors of $N$. Each distribution corresponds to a unique pair $(a, b)$ such that $a \times b = N$ and $\gcd(a, b) = 1$.The number of such pairs is $2^{\omega(N)}$, where $\omega(N)$ is the number of distinct prime factors of $N$.Since 20! is not a perfect square, for every pair $(a, b)$ such that $a \times b = 20!$ and $\gcd(a, b) = 1$, we have $a \neq b$. The pairs can be grouped into $(a, b)$ and $(b, a)$, where $a \neq b$.The number of pairs with $a < b$ is equal to the number of pairs with $b < a$.The total number of pairs is $2^{\omega(20!)}$. The number of pairs with $a < b$ is $\frac{1}{2} \times 2^{\omega(20!)} = 2^{\omega(20!)-1}$.The number of distinct prime factors of 20! is $\omega(20!) = 8$.The number of pairs $(a, b)$ such that $a \times b = 20!$, $\gcd(a, b) = 1$, and $a < b$ is $2^{8-1} = 2^7 = 128$.Each such pair $(a, b)$ corresponds to a unique rational number $\frac{a}{b}$ between 0 and 1 in lowest terms, with the product of the numerator and denominator equal to 20!.Final Answer: The final answer is $\boxed{128}$"}

Solution:
{ ReasoningStep: "Let the rational number between 0 and 1 be $r = \frac{a}{b}$ in lowest terms with $\gcd(a, b) = 1$, $0 < a < b$, and $a \times b = 20!$".

Tag:"Reducing a fraction to lowest terms",

Question:"In the context of finding rational numbers $r = \frac{a}{b}$ in lowest terms with $a \cdot b = 20!$, suppose you start with the unreduced fraction $\frac{7}{20!/7}$. Reduce this fraction to lowest terms.",

Solution:{

ReasoningStep: "We have the fraction $F = \frac{7}{20!/7}$. Since 20! contains $7^2$ (because $\lfloor 20/7 \rfloor = 2$), the number $20!/7$ is still divisible by 7.",

Tag:"Exponent of a prime in a factorial (Legendre's formula)",

Question:"In the context of reducing the fraction 7/(20!/7), use Legendre's formula to find the exponent of the prime 7 in the prime factorization of 20!.",

Solution:" 1. Legendre's formula states that the exponent of a prime p in n! is given by the sum of $\lfloor n/p^k \rfloor$ for $k = 1, 2, .. p^k > n$.
2. Here, n=20 and p=7:
- For k=1: $\lfloor 20/7 \rfloor = 2$
- For k=2: $\lfloor 20/49 \rfloor = 0$
- Higher powers of 7 exceed 20, so their terms are zero.
3. Summing these gives $2 + 0 = 2$."
},

*Table 7.* **Summary of Cluster Tags**. Sample count represents the number of mathematical problems associated with each cluster tag. Keywords are derived from the tags within each cluster.

| Domain | #questions | Keywords |
| --- | --- | --- |
| Equation Manipulation | 17 | Equation Manipulation, Equation Simplification, Equation Solving, Equation Subtraction, Linear Equation, Linear Equation in Two Variables, Linear Equations, Polynomial Equation Solving, Polynomial Simplification, Quadratic Equation, Rational Equation, Rational Equation Solving, Rearranging Equations, Simplifying Expressions, Simultaneous Equations, Simultaneous Equations Solving, Solving Rational Equations, Substitution, Subtraction, Variable substitution |
| Elimination Method | 14 | Elimination Method, Equation Solving: Isolating Variables, Linear Equation Solving, Solving Linear Equations, Substitution Method |
| Addition | 12 | Addition, Addition of Integers, Arithmetic Addition, Arithmetic Operations, Column Addition, Digit Sum, Fraction Addition, Integer Addition, Multiplication, Place Value Addition, Summation |
| Divisibility | 11 | Divisibility, Divisibility Rules, Division Property of Equality, Polynomial Division |
| Clearing Fractions | 10 | Clearing Fractions, Common Denominator Calculation, Fraction Multiplication, Fraction Simplification, Fraction Subtraction, Partial Fraction Decomposition, Reciprocal Calculation, Simplifying Fractions, Subtracting Fractions with Common Denominator |
| Arithmetic Sequence | 7 | Arithmetic Sequence, Arithmetic Sequence Formula, Arithmetic Subtraction, Counting Integers in an Arithmetic Sequence, Modular Arithmetic, Sum of a Sequence, Sum of an Arithmetic Series |
| Factoring Polynomials | 6 | Factoring Polynomials, Factoring Quadratic Equations, Factoring by Grouping, Factoring by grouping, Polynomial Expansion, Prime Factorization |
| Binomial Coefficient | 6 | Binomial Coefficient, Binomial Coefficient Calculation, Combination Formula, Combinations Calculation, Combinatorial Counting, Sum of coefficients |
| Complementary Counting | 6 | Complementary Counting, Counting Principle, Counting Principles, Counting Rows, Inclusion-Exclusion Principle |
| Combining Like Terms | 6 | Combining Like Terms |
| Cross-Multiplication | 5 | Cross-Multiplication, Multiplication Principle, Prime Multiplication, Scalar Multiplication |
| Division | 5 | Division, Division of Equations, Division of constants, Long Division |
| Absolute Value | 4 | Absolute Value, Absolute Value Calculation, Absolute Value Equation, Absolute Value Equation Solving, Absolute Value Equations, Magnitude of a Complex Number, Magnitude of a Vector |
| Isolating Variables | 4 | Isolating Variables, Isolating the Variable |
| GCD Calculation | 3 | GCD Calculation, GCD Property |
| Cauchy-Schwarz Inequality | 3 | Cauchy-Schwarz Inequality, Compound Inequalities, Inequalities, Inequality Manipulation, Linear Inequality Simplification |
| Altitude in Triangles | 3 | Altitude in Triangles, Altitude of a Triangle, Similar Triangles, Triangle Construction |
| Stars and Bars | 3 | Stars and Bars, Stars and Bars Method |
| Exponentiation | 3 | Exponentiation, Logarithm Power Rule, Modular Exponentiation |
| Square Root Calculation | 3 | Square Root Calculation, Squaring a number, Squaring both sides |
| Order of Operations | 2 | Order of Operations, Order of an Element |
| Identifying Parallel Lines | 2 | Identifying Parallel Lines, Line Intersection, Parallel Lines in Polygons, Slope of Parallel and Perpendicular Lines, Slope of Perpendicular Lines |
| Perpendicular Slopes | 2 | Perpendicular Slopes, Slope of a Line, Vertical Tangent Line |
| Quadratic Equation Identification | 2 | Quadratic Equation Identification, Quadratic Formula |
| Combinatorial Placement | 2 | Combinatorial Placement |
| Solving Linear Inequalities | 2 | Solving Linear Inequalities, System of Linear Equations |
| Cyclic Distance Calculation | 2 | Cyclic Distance Calculation, Distance Formula |
| Euler's Relation for Triangle Centers | 2 | Euler's Relation for Triangle Centers, Euler's Theorem |
| Arc Length Calculation | 2 | Arc Length Calculation, Perimeter Calculation |
| Angle Bisector Theorem | 2 | Angle Bisector Theorem, Perpendicular Bisector of a Chord |
| Bayes' Theorem | 1 | Bayes' Theorem, Conditional Probability |
| Distributive Property | 1 | Distributive Property |
| Floor Function | 1 | Floor Function |
| Finding Zeros of a Function | 1 | Finding Zeros of a Function |
| Factorial Calculation | 1 | Factorial Calculation |
| Pigeonhole Principle | 1 | Pigeonhole Principle |
| Conclusion Verification | 1 | Conclusion Verification |
| Change of Base Formula | 1 | Change of Base Formula |
| Logarithm Properties | 1 | Logarithm Properties, Logarithmic Identity, Logarithmic Properties, Logarithmic Reciprocity |
| Summation of a finite set | 1 | Summation of a finite set |
| Complex Number Magnitude | 1 | Complex Number Magnitude, Complex Number Scaling, Dot Product Magnitude, Modulus of a Complex Number, Vector Magnitude Calculation |
| Place Value | 1 | Place Value |
| Chinese Remainder Theorem | 1 | Chinese Remainder Theorem |
| Lifting The Exponent Lemma (LTE) | 1 | Lifting The Exponent Lemma (LTE) |
| Order of an Element in Modular Arithmetic | 1 | Order of an Element in Modular Arithmetic, Order of an element modulo p |
| Primitive Root Calculation | 1 | Primitive Root Calculation |
| $p$-adic Valuation | 1 | $p$-adic Valuation, $v_p$ (p-adic valuation) |
| Diagonal of a Rectangular Prism Calculation | 1 | Diagonal of a Rectangular Prism Calculation |
| Pythagorean Theorem | 1 | Pythagorean Theorem, Pythagorean Theorem in 3D |
| Interior Angle of a Regular Polygon | 1 | Interior Angle of a Regular Polygon |
| Pairing Elements | 1 | Pairing Elements |
| Power of a Point Theorem | 1 | Power of a Point Theorem |
| Bipartite Graph Matching | 1 | Bipartite Graph Matching |
| Permutation Calculation | 1 | Permutation Calculation |
| Prime Identification | 1 | Prime Identification, Prime Number Identification, Prime Number Multiplication |
| Triangle Perimeter Calculation | 1 | Triangle Perimeter Calculation |
| Triangle Side Classification | 1 | Triangle Side Classification |
| Counterexample | 1 | Counterexample |
| Mode Calculation | 1 | Mode Calculation, Mode Identification |
| Sorting Numbers | 1 | Sorting Numbers |
| Area of a Circle | 1 | Area of a Circle, Area of a Circle Calculation |
| Cross-Section of a Sphere | 1 | Cross-Section of a Sphere, Cross-sections of spheres |
| LCM Calculation | 1 | LCM Calculation |
| Proportion | 1 | Proportion, Proportion Solving, Ratio and Proportion |
| Radius of a sphere | 1 | Radius of a sphere |
| Volume of a Tetrahedron | 1 | Volume of a Tetrahedron |
| Newton's Sums | 1 | Newton's Sums |
| Vieta's Formulas | 1 | Vieta's Formulas, Vieta's formulas |

*Table 8.* **Summary of Cluster Tags (Sample Count ≥ 3)**. Sample count represents the number of mathematical problems associated with each cluster tag. Keywords are derived from the tags within each cluster.

| Domain | #questions | Keywords |
| --- | --- | --- |
| Combinatorial Probability | 100 | Combinatorial Probability, Counting & Probability |
| Basic Counting Principle | 46 | Basic Counting Principle, Counting Principle, Counting Principles, Fundamental Counting Principle, Fundamental Principle of Counting, Multiplication Principle of Counting |
| Combination Calculation | 36 | Combination Calculation, Combination Enumeration, Combination Formula, Combination Selection, Combination Subtraction, Combination and Permutation Calculation, Combinations, Combinations Calculation, Combinations Formula, Counting Combinations |
| Factorial | 31 | Factorial, Factorial Calculation, Factorial Division, Factorial Expansion, Factorial Manipulation, Factorial Multiplication, Factorial Properties, Factorial Simplification, Factorial simplification |
| Combinatorial Counting | 30 | Combinatorial Counting, Combinatorial Enumeration, Combinatorial Exclusion, Combinatorial Reasoning, Combinatorics, Counting, Counting Arrangements, Counting Multiples, Counting Subsets |
| Binomial Coefficient | 29 | Binomial Coefficient, Binomial Coefficient Calculation, Binomial Coefficient Formula, Binomial Coefficient Multiplication, Binomial Coefficient Simplification, Binomial Coefficients, Binomial Coefficients Calculation, Binomial Expansion, Binomial Probability Formula, Binomial Theorem |
| Fraction Conversion | 28 | Fraction Conversion, Fraction Identification, Fraction Multiplication, Fraction Subtraction, Multiplication of Fractions, Percentage to Fraction Conversion, Simplifying Fractions |
| Linear Equation Evaluation | 26 | Linear Equation Evaluation, Linear Equation Solving, Linear Expression Evaluation, Solving Linear Equations |
| Division Simplification | 24 | Division Simplification, Division of Fractions, Fraction Division, Fraction Multiplication and Simplification, Fraction Simplification, Ratio Simplification, Simplifying Ratios |
| Addition | 23 | Addition, Addition of Integers, Addition of integers, Arithmetic Addition, Integer Addition |
| Arithmetic Operations | 21 | Arithmetic Operations, Basic Arithmetic Subtraction, Multiplication, Multiplication of Integers, Multiplication of integers |
| Division Property of Equality | 21 | Division Property of Equality |
| Exponent Simplification | 21 | Exponent Simplification, Exponentiation |
| Basic Probability Calculation | 19 | Basic Probability Calculation, Conditional Probability Calculation, Probability, Probability Calculation |
| Equation Substitution | 18 | Equation Substitution, Substitution, Substitution Method, Variable Substitution |
| Circular Permutations | 17 | Circular Permutations, Cyclic Permutations, Permutation, Permutations |
| Set Subtraction | 13 | Set Subtraction, Subtraction |
| Division | 12 | Division, Division Algorithm, Long Division |
| GCD Calculation | 12 | GCD Calculation |
| Independent Probability Multiplication | 11 | Independent Probability Multiplication, Multiplication Rule for Independent Events, Multiplication Rule for Probabilities, Probability Multiplication Rule |
| Isolating Variables | 10 | Isolating Variables, Isolating the variable |
| Combinations with Repetition | 9 | Combinations with Repetition, Permutations and Combinations, Permutations with Repetition |
| Counting Exclusion | 9 | Counting Exclusion, Exclusion Principle, Inclusion-Exclusion Principle |
| Counting and Summation | 8 | Counting and Summation, Summation |
| Adding Fractions | 8 | Adding Fractions, Adding Fractions with Like Denominators, Adding Fractions with Unlike Denominators, Addition of Fractions, Arithmetic with Fractions, Common Denominator Addition, Fraction Addition, Multiplying Fractions |
| Prime Factorization | 8 | Prime Factorization |
| Combining Like Terms | 7 | Combining Like Terms |
| Place Value | 7 | Place Value, Place Value Identification |
| Arithmetic Sequence | 6 | Arithmetic Sequence, Arithmetic Sequence Formula, Arithmetic Sequence Identification, Arithmetic Sequence Sum, Arithmetic Sequence Sum Formula, Arithmetic Sequence Summation, Arithmetic Sequences, Arithmetic Sequences Counting, Counting Terms in an Arithmetic Sequence |
| Equation Simplification | 6 | Equation Simplification, Linear Equation Simplification, Polynomial Simplification |
| Conditional Probability | 6 | Conditional Probability |
| Permutation with Restrictions | 6 | Permutation with Restrictions, Permutations with Restrictions, Permutations with restrictions |
| Divisibility Rule for 3 | 6 | Divisibility Rule for 3, Divisibility Rules |
| Factoring by grouping | 6 | Factoring by grouping, Factorization |
| Complement Rule | 5 | Complement Rule, Complement Rule in Probability |
| Multiplication Principle | 4 | Multiplication Principle |
| Arithmetic Series Formula | 4 | Arithmetic Series Formula, Arithmetic Series Sum Formula, Arithmetic Series Summation, Arithmetic Sum Calculation, Summation of Arithmetic Series |
| Order of Operations | 4 | Order of Operations |
| Equation Balancing | 3 | Equation Balancing |
| Binomial Probability | 3 | Binomial Probability |
| Counting Integers | 3 | Counting Integers, Counting Integers in a Range |
| Power Set Calculation | 3 | Power Set Calculation |
| Discriminant Calculation | 3 | Discriminant Calculation |
| Identifying Coefficients in a Quadratic Equation | 3 | Identifying Coefficients in a Quadratic Equation, Quadratic Coefficients Identification |
| Complementary Counting | 3 | Complementary Counting |
| Distributive Property | 3 | Distributive Property |
| Combination Symmetry | 3 | Combination Symmetry, Combinatorial Symmetry, Symmetry Counting |
| Area Calculation | 3 | Area Calculation, Area Calculation of a Square, Area Ratios, Area of a Rectangle Calculation, Area of a Square Calculation, Area of a Triangle Calculation |
| Area of a Right Triangle | 3 | Area of a Right Triangle, Area of a Triangle, Triangle Area Formula |
| Counting Outcomes | 3 | Counting Outcomes, Enumerating Outcomes |
| Independent Events | 3 | Independent Events, Independent Events Probability, Independent Events Probability Calculation, Probability of Independent Events |
| Scalar Multiplication | 3 | Scalar Multiplication |
| Case Analysis | 3 | Case Analysis |
| Common Denominator Calculation | 3 | Common Denominator Calculation, Common Denominator Conversion, Finding a Common Denominator, Subtracting Fractions with Common Denominator, Subtracting Fractions with Common Denominators |
| Finding the Least Common Multiple (LCM) | 3 | Finding the Least Common Multiple (LCM), LCM Calculation, Least Common Multiple (LCM) Calculation |
| Factoring Common Factor | 3 | Factoring Common Factor, Factoring Out Common Factors, Finding Factors |
| Counting Even Numbers | 3 | Counting Even Numbers, Counting Odd Numbers, Even Numbers Identification, Even and Odd Numbers, Identifying Even Numbers |
| Modular Arithmetic | 3 | Modular Arithmetic, Modulo Operation |

*Table 9.* **Summary of Cluster Tags (Sample Count < 3)**. Sample count represents the number of mathematical problems associated with each cluster tag. Keywords are derived from the tags within each cluster.

| Domain | #questions | Keywords |
|---|---|---|
| Probability Distribution | 2 | Probability Distribution |
| Range of Sums for Dice Rolls | 2 | Range of Sums for Dice Rolls, Sum of Two Dice Rolls |
| Uniform Probability Distribution | 2 | Uniform Probability Distribution |
| Properties of Platonic Solids | 2 | Properties of Platonic Solids, Symmetry of Platonic Solids |
| Division of Constants | 2 | Division of Constants, Division of constants |
| Permutations of Multisets | 2 | Permutations of Multisets |
| Digit Fixation in Positional Notation | 2 | Digit Fixation in Positional Notation, Digit Placement |
| Multiples Identification | 2 | Multiples Identification |
| Distance Formula | 2 | Distance Formula, Horizontal Distance Calculation |
| Graphing Inequalities | 2 | Graphing Inequalities, Graphing Linear Inequalities, Linear Inequality Graphing |
| Intersection of Lines and Curves | 2 | Intersection of Lines and Curves, Line Intersection |
| Isosceles Right Triangle | 2 | Isosceles Right Triangle, Isosceles Triangle Properties |
| Probability of Combined Events | 2 | Probability of Combined Events, Probability of a Single Event |
| Combinatorial Selection | 2 | Combinatorial Selection, Subset Selection |
| Subset Identification | 2 | Subset Identification |
| Complementary Probability | 2 | Complementary Probability |
| Probability Addition Rule | 2 | Probability Addition Rule, Total Probability Rule |
| Factor Pairing | 2 | Factor Pairing, Factor Pairs Identification |
| Finding Multiples | 2 | Finding Multiples, Multiples of a Number |
| Prime Identification | 2 | Prime Identification, Prime Number Identification, Prime and Composite Numbers Identification |
| Expected Value Calculation | 2 | Expected Value Calculation |
| Addition and Subtraction Properties of Equality | 2 | Addition and Subtraction Properties of Equality |
| Long Multiplication | 2 | Long Multiplication, Multiplication of Large Numbers |
| Geometric Series | 2 | Geometric Series, Geometric Series Formula, Geometric Series Identification, Geometric Series Sum Formula, Geometric Series Summation, Infinite Geometric Series Formula, Sum of Infinite Geometric Series |
| Pascal's Triangle Construction | 2 | Pascal's Triangle Construction, Pascal's Triangle Row Sum |
| Independent Probability | 2 | Independent Probability |
| Exponentiation of Fractions | 2 | Exponentiation of Fractions |
| Simplifying Rational Expressions | 2 | Simplifying Rational Expressions |
| Pascal's Identity | 2 | Pascal's Identity, Pascal's Triangle |
| Summation of Series | 2 | Summation of Series, Summation of a Sequence |
| Intersection of Sets | 2 | Intersection of Sets, Set Intersection |
| Set Union | 2 | Set Union, Set Union Cardinality |
| Percentage Calculation | 2 | Percentage Calculation, Percentage Conversion, Percentage to Decimal Conversion |
| Symmetry in Probability | 2 | Symmetry in Probability |
| Counting Grid Positions | 2 | Counting Grid Positions, Counting Rectangles in a Grid, Counting Squares in a Grid, Counting Subsets in a Grid |
| Parity | 2 | Parity |
| Recurrence Relation | 2 | Recurrence Relation, Recurrence Relations |
| Block Permutation | 2 | Block Permutation |
| Digit Pairing for Sum | 2 | Digit Pairing for Sum, Digit Sum Calculation, Pairing Numbers for a Fixed Sum |
| Permutation Calculation | 2 | Permutation Calculation |
| Cyclic Number Patterns | 2 | Cyclic Number Patterns, Cyclic Sequences |
| Bipartite Graph | 2 | Bipartite Graph, Bipartite Graph Coloring |
| Digit Constraints | 2 | Digit Constraints, Digit Restriction, Digit Sum Constraints, Single-digit constraint |
| Counting Leap Years | 1 | Counting Leap Years, Leap Year Calculation |
| Minimum Value Calculation | 1 | Minimum Value Calculation |
| Pigeonhole Principle | 1 | Pigeonhole Principle |
| Range Calculation | 1 | Range Calculation |
| Burnside's Lemma | 1 | Burnside's Lemma |
| Polyhedron Properties | 1 | Polyhedron Properties, Properties of Polyhedra |
| Rotational Symmetry | 1 | Rotational Symmetry |
| Rotational Symmetry of Polyhedra | 1 | Rotational Symmetry of Polyhedra, Symmetry in Polyhedra |
| Slope Calculation | 1 | Slope Calculation |
| Pair Counting | 1 | Pair Counting |
| Pairwise Sum Calculation | 1 | Pairwise Sum Calculation |
| Division of Even Numbers | 1 | Division of Even Numbers |
| Frequency Distribution | 1 | Frequency Distribution |
| Conditional Statements | 1 | Conditional Statements |
| Counting Intervals | 1 | Counting Intervals |
| Period Calculation | 1 | Period Calculation |
| Time Interval Calculation | 1 | Time Interval Calculation |
| Unit Conversion | 1 | Unit Conversion |
| Factoring Quadratic Expressions | 1 | Factoring Quadratic Expressions |
| Simultaneous Equations | 1 | Simultaneous Equations |
| Counting Cyclic Quadrilaterals with Integer Sides | 1 | Counting Cyclic Quadrilaterals with Integer Sides |
| Counting Rectangles | 1 | Counting Rectangles |
| Hockey Stick Identity | 1 | Hockey Stick Identity |
| Perimeter Calculation | 1 | Perimeter Calculation |
| Properties of Quadrilaterals | 1 | Properties of Quadrilaterals, Symmetry in Quadrilaterals |
| Properties of a Square | 1 | Properties of a Square |
| Stars and Bars Method | 1 | Stars and Bars Method |
| Triangle Inequality | 1 | Triangle Inequality |
| Distance Comparison | 1 | Distance Comparison |
| Absolute Value Simplification | 1 | Absolute Value Simplification |
| Subset Definition | 1 | Subset Definition |
| Equivalent Fractions | 1 | Equivalent Fractions |
| Cube Root Estimation | 1 | Cube Root Estimation |
| Inequality Comparison | 1 | Inequality Comparison |
| Sequential Multiplication | 1 | Sequential Multiplication |
| Angle Measurement in Radians | 1 | Angle Measurement in Radians, Radian-Degree Conversion |
| Arc Length Calculation | 1 | Arc Length Calculation, Arc Length Formula |
| Arc Measure | 1 | Arc Measure |
| Central Angle Theorem | 1 | Central Angle Theorem |
| Circumference of a Circle | 1 | Circumference of a Circle |
| Commutative Property of Addition | 1 | Commutative Property of Addition |
| Factorial Decomposition | 1 | Factorial Decomposition |
| Matrix Indexing | 1 | Matrix Indexing |
| Probability with Replacement | 1 | Probability with Replacement |

{

ReasoningStep:"Compute gcd(7, 20!/7). Because 20! contains $7^2$, dividing by 7 leaves one factor of 7, so gcd(7, 20!/7) = 7.",

Tag:"GCD calculation using prime factorization",

Question:"In reducing the fraction 7 one over (20! divided by 7) to lowest terms, compute gcd(7, 20! / 7) by using prime factorization.",

Solution:"1. We need the exponent of the prime 7 in 20!.

2. Use the formula for exponent of a prime p in n!: $v_7(20!) = \lfloor 20/7 \rfloor + \lfloor 20/49 \rfloor = 2 + 0 = 2$.

3. Thus 20! has a factor $7^2$.

4. When we divide 20! by 7, the exponent of 7 in (20!/7) is 2  1 = 1.

5. So (20!/7) contains exactly one factor of 7, and the number 7 also contains one factor of 7.

6. Therefore gcd(7, 20!/7) = $7^1$ = 7."

}

{

ReasoningStep:"Divide numerator and denominator by 7: Numerator: $7 \div 7 = 1$. Denominator: $(20!/7) \div 7 = 20!/(7^2) = 20!/49$.",

Tag:"Cancelling common factors in a fraction",

Question:"In the context of the original problem, you encounter the fraction $\frac{7}{20!/7}$. By cancelling the common factor 7 in the numerator and denominator, what is the resulting simplified denominator?",

Solution:"1. Start with the fraction $\frac{7}{20!/7}$. 2. Observe that both the numerator and denominator contain the factor 7. 3. Divide numerator and denominator by 7: - New numerator: $7 \div 7 = 1$. - New denominator: $(20!/7) \div 7 = 20!/(7^2) = 20!/49$. 4. Thus, after cancellation, the fraction becomes $\frac{1}{20!/49}$. 5. The question asks for the simplified denominator, which is 20!/49."

},

{

ReasoningStep:"Hence the fraction in lowest terms is $\frac{1}{20!/49}$.",

Tag:"Reducing fractions to lowest terms",

Question:"Reduce the fraction 7 divided by (20! divided by 7) to lowest terms.",

Solution:"1. Start with the unreduced fraction: 7 / (20! / 7).

2. Rewrite as a single fraction: $7 \times 7 / 20! = 49 / 20!$.

3. Observe that 20! contains the factor $7^2 = 49$, $so gcd(49, 20!) = 49$.

4. Divide numerator and denominator by 49: ● Numerator: $49 \div 49 = 1$ ● Denominator: $20! \div 495$. Hence the fraction in lowest terms is 1 / (20! / 49).

6. Compute 20! = 2 432 902 008 176 640 000, then divide by 49 to get 49 651 061 391 360 000.

7. Final reduced form is 1 / 49651061391360000.",],

{

ReasoningStep:"Set $N = 20!$. Its prime factorization has $k = 8$ distinct primes (2, 3, 5, 7, 11, 13, 17, 19).",

Tag:"Prime Factorization",

Question:"How many distinct prime numbers appear in the prime factorization of 20!?",

Solution:

{ ReasoningStep:"By definition, 20! = 1×2×3×...×20, so every prime p  20 divides 20!.",

Tag:"Definition of factorial",

Question:"In the definition of factorial, 20! is written as the product 1×2×3×...×20. How many integers are multiplied together in this product?",

Solution:"By definition, n! = 1×2×3×...×n, which multiplies all integers from 1 up to n. For n = 20, the integers are 1, 2, 3, ..., 20. Counting these gives 20 numbers being multiplied.",

{ ReasoningStep:"The primes less than or equal to 20 are 2, 3, 5, 7, 11, 13, 17, and 19, and there are no others in that range.",

Tag:"Primality Testing",

Question:"Among the integers from 1 through 20 inclusive, how many are prime numbers?",

Solution:"1. By definition, a prime number is an integer greater than 1 that has no positive divisors other than 1 and itself.

2. We test each integer from 1 to 20: - 1 is not prime by definition. - 2 is divisible only by 1 and 2 → prime. - 3 is divisible only by 1 and 3 → prime. - 4 is divisible by 1, 2, 4 → not prime. - 5 is divisible only by 1 and 5 → prime. - 6 is divisible by 1, 2, 3, 6 → not prime. - 7 is divisible only by 1 and 7 → prime. - 8 is divisible by 1, 2, 4, 8 → not prime. - 9 is divisible by 1, 3, 9 → not prime. - 10 is divisible by 1, 2, 5, 10 → not prime. - 11 is divisible only by 1 and 11 → prime. - 12 is divisible by 1, 2, 3, 4, 6, 12 → not prime. - 13 is divisible only by 1 and 13 → prime. - 14 is divisible by 1, 2, 7, 14 → not prime. - 15 is divisible by 1, 3, 5, 15 → not prime. - 16 is divisible by 1, 2, 4, 8, 16 → not prime. - 17 is divisible only by 1 and 17 → prime. - 18 is divisible by 1, 2, 3, 6, 9, 18 → not prime. - 19 is divisible only by 1 and 19 → prime. - 20 is divisible by 1, 2, 4, 5, 10, 20 → not prime.

3. The primes in this range are 2, 3, 5, 7, 11, 13, 17, and 19.

4. Counting them gives a total of 8 primes.",

{ ReasoningStep:"Therefore, the prime factorization of 20! includes exactly these 8 distinct primes.",

Tag:"Prime Factorization",

Question:"When prime factorizing 20!, we include every prime number that is less than or equal to 20. How many distinct prime numbers appear in the prime factorization of 20! ?",

Solution:"1. A prime divides 20! exactly if it is  20, since 20! is the product of all integers from 1 to 20.

2. The primes  20 are 2, 3, 5, 7, 11, 13, 17, and 19.

3. Counting these gives 8 distinct primes in the factorization of 20!." },

{ ReasoningStep:"Since gcd($a, b$) = 1, each prime power $p_i^{e_i}$ of $N$ must go entirely to $a$ or $b$, giving $2^k$ pairs ($a, b$).",

Tag:"Fundamental Theorem of Arithmetic",

Question:"Let N be 20! and write N in its prime factorization N = $p_1^{e_1} \cdot p_2^{e_2} \cdot \ldots \cdot p_k^{e_k}$.

Using the Fundamental Theorem of Arithmetic, in how many ways can you choose a pair of positive integers (a, b) such that ab = N and gcd(a, b) = 1 by assigning each prime power $p_i^{e_i}$ entirely to either a or b?",

Solution:{ReasoningStep:"By the Fundamental Theorem of Arithmetic, 20! factors into prime powers corresponding to each prime p  20.",

Tag:"Prime Factorization",

Question:"In the prime factorization of 20!, what is the exponent of the prime 2?",

Solution:"1. By Legendre's formula, the exponent of a prime p in n! is $\lfloor n/p \rfloor + \lfloor n/p^2 \rfloor + \lfloor n/p^3 \rfloor +$ … until the terms are zero.

2. For p = 2 and n = 20: $\lfloor 20/2 \rfloor = 10 \lfloor 20/4 \rfloor = 5 \lfloor 20/8 \rfloor = 2 \lfloor 20/16 \rfloor = 1 \lfloor 20/32 \rfloor = 0$ (and all higher powers give zero)

3. Sum these: $10 + 5 + 2 + 1 = 18$.",

},

{ReasoningStep:"The primes  20 are 2, 3, 5, 7, 11, 13, 17, 19, so there are k = 8 distinct prime powers in the factorization of 20!.",

Tag:"Listing primes up to a given limit",

Question:"In the prime factorization of 20!, how many distinct prime bases (i.e., prime numbers  20) appear?",

Solution:"

1. To find the distinct prime factors of 20!, we list all prime numbers that are less than or equal to 20.

2. A prime number is a number greater than 1 with no positive divisors other than 1 and itself.

3. Checking the integers from 2 up to 20, the primes are: 2, 3, 5, 7, 11, 13, 17, 194. Count these primes: there are 8 of them.",

},

{ReasoningStep:"To form a pair (a, b) with ab = N and gcd(a, b) = 1, each prime power $p_i^{e_i}$ must go entirely to a or entirely to b.",

Tag:"Fundamental Theorem of Arithmetic",

Question:"Let N = 20! and write its prime factorization as $N = p_1^{e_1} \cdot p_2^{e_2} \cdot \ldots \cdot p_k^{e_k}$. By the Fundamental Theorem of Arithmetic, to form pairs of positive integers (a, b) with ab = N and gcd(a, b) = 1, each prime power $p_i^{e_i}$ must go entirely to a or entirely to b. For a single prime power $p_i^{e_i}$, in how many ways can you assign it to either a or b so that gcd(a, b) = 1?",

Solution:" 1. The Fundamental Theorem of Arithmetic ensures N's prime factorization is unique.

2. To keep gcd(a, b) = 1, you cannot split $p_i^{e_i}$ between a and b.

3. Therefore, for that prime power, you have exactly two choices: - Assign $p_i^{e_i}$ to a (so exponent in a is $e_i$, in b is 0), - or assign $p_i^{e_i}$ to b (so exponent in a is 0, in b is $e_i$).

4. No other distributions are allowed without violating gcd(a, b) = 1.",

},

{ReasoningStep:"Each of the k = 8 prime powers has 2 choices, so by the multiplication principle there are $2^k = 2^8$ ways.",

Tag:"Fundamental Counting Principle",

Question:"Suppose the prime factorization of 20! contains 8 distinct prime powers. If you assign each prime power entirely to either integer a or integer b, in how many ways can you make these assignments by the Fundamental Counting Principle?",

Solution:"1. There are 8 prime powers to assign.

2. Each prime power has 2 independent choices: assign it to a or to b.

3. By the Fundamental Counting Principle, multiply the number of choices for each prime power: Total ways = 2•2•…•2(8 times) = $2^8$ = 256.",

}],

},

{ReasoningStep:"As 20! is not a perfect square, $a \neq b$, so half the pairs have $a < b$, yielding $2^{k-1} = 2^7 = 128$.",

Tag:"Divisor Pairing Principle",

Question:"Using the Divisor Pairing Principle, in how many ways can we write 20! as a product ab of two positive integers with gcd(a,b)=1 and a<b?",

Solution:

{ReasoningStep:"The prime factorization of 20! involves exactly k = 8 distinct primes (2, 3, 5, 7, 11, 13, 17, 19).",

Tag:"Prime factorization",

Question:"In the prime factorization of 20!, how many distinct prime factors does it contain?",

Solution:"1. By definition, $20! = 1 \cdot 2 \cdot 3 \cdot \ldots \cdot 20$.

2. Every prime p  20 divides one of the factors in the product.

3. The prime numbers less than or equal to 20 are 2, 3, 5, 7, 11, 13, 17, and 19.

4. There are 8 such primes.",

},

{ReasoningStep:"To have ab = 20! and gcd(a, b) = 1, each prime's entire power in 20! must go either to a or to b.",

Tag:"Unique Prime Factorization",

Question:"In the prime factorization of 20!, what is the exponent of the prime 3?",

Solution:"1. By unique prime factorization, the exponent of a prime p in n! is given by summing $\lfloor n/p^k \rfloor \, for \, k \geq 1 \, until \, p^k > n$.

2. For p=3 and n=20: $- \lfloor 20/3 \rfloor = 6 \lambda \lfloor 20/9 \rfloor = 2 \lambda \lfloor 20/27 \rfloor = 0 \, (stop\,here)$

3. Sum of these is $6 + 2 = 8$.",

},

{ReasoningStep:"Therefore there are $2^k = 2^8 = 256$ unordered assignments of prime-powers to (a, b).",

Tag:"Fundamental Counting Principle",

Question:"The prime factorization of 20! involves 8 distinct prime-power factors. Suppose each entire prime-power factor must be assigned either to integer a or to integer b. Using the Fundamental Counting Principle, in how many ways can these 8 prime powers be distributed between a and b?",

Solution:"1. There are 8 distinct prime-power factors in 20! (for primes 2, 3, 5, 7, 11, 13, 17, 19).

2. For each prime-power factor, we have exactly 2 choices: assign it to a or assign it to b.

3. By the Fundamental Counting Principle, the total number of ways to make all choices is $2 \times 2 \times \ldots \times 2$ (8 factors) $= 2^8$.

4. Compute $2^8 = 256$.",

},

{ReasoningStep:"Since 20! is not a perfect square, no assignment yields a = b, so exactly half of these yield a < b, giving 256/2 = 128.",

Tag:"Symmetry argument in combinatorial counting",

Question:"Suppose there are 256 ordered pairs of positive integers (a,b) such that ab = 20! and gcd(a,b) = 1. Using a symmetry argument, how many of these pairs satisfy a < b?",

Solution:"1. We are given that there are 256 ordered coprime factor pairs (a,b) with ab = 20!.

2. For each ordered pair (a,b), there is a corresponding "swapped" pair (b,a).

3. Because 20! is not a perfect square, no pair has a = b; every pair is distinct from its swap.

4. Thus the 256 ordered pairs split evenly into two groups: those with a < b and those with a > b.

5. By symmetry, the number with a < b is half of 256, namely 256/2 = 128.",}, }

