# OpenReview forum: "Learning to Solve Complex Problems via Dataset Decomposition"
_NeurIPS.cc/2025/Conference — NeurIPS 2025 poster_

### Official Review · Reviewer_nBar · 2025-06-04

**Clarity:** 3
**Significance:** 2
**Originality:** 3
**Rating:** 4
**Confidence:** 4

**Summary:**

This paper introduces a curriculum learning strategy where a "teacher" Large Language Model (LLM) recursively decomposes complex mathematical problems into simpler, hierarchically structured sub-problems. Each sub-problem is tagged with a concept and verified. A novel scoring system quantifies the difficulty of these sub-problems based on their structural complexity (number of sub-steps) and conceptual depth (derived from a "Concept Dependency Graph" that maps prerequisite skills). Smaller "student" LLMs are then trained on this decomposed dataset in stages of increasing difficulty. Experiments on MATH and AIME datasets show that this approach significantly improves the performance and generalization of student models compared to standard training and other data augmentation methods. The method's success hinges on the teacher model's capability.

**Questions:**

1. This method involves a more complex procedure than direct distillation and underperforms compared to distillation using a reasoning model. What advantages does it offer over directly distilling with a reasoning model?

2. Is this method effective for reasoning models? Can it further enhance their performance?

3. How dependent is this method on the capability of the Teacher Model? Can it still achieve good results when the Teacher Model itself performs poorly?

**Ethical Concerns:**

["NO or VERY MINOR ethics concerns only"]

**Final Justification:**

This paper explores an interesting direction: leveraging a strong teacher model to decompose datasets in a way that facilitates learning for smaller models. While the authors present promising results and have addressed some concerns in the rebuttal, I still have the following reservations:

Generalizability remains unclear. The paper does not evaluate whether the method improves performance for reasoning-capable student models, nor does it test on out-of-distribution datasets (e.g., training on a subset of MATH and evaluating on Math500, or training on AIME 2024 and testing on AIME 2025).

Effectiveness at scale is uncertain. Although the authors show improvements over direct distillation in small-scale experiments, their best-performing model underperforms compared to existing strong baselines such as DeepSeek-R1-Distill. It remains unclear whether the proposed method can match or surpass such baselines in large-scale training settings.

Overall, I appreciate the novelty and the direction of the work, but these concerns limit my enthusiasm for a strong recommendation.

**Limitations:**

yes

**Quality:**

2

**Strengths And Weaknesses:**

Strengths:

1. Novel Methodology: The paper proposes a novel "reverse curriculum generation" approach that recursively decomposes complex problems into simpler, learnable components using a teacher-student framework. This hierarchical decomposition is a key innovation.

2. Sophisticated Difficulty Scoring: It introduces a unique system for measuring data difficulty based on both "structural complexity" (number of sub-problems) and "conceptual depth" (derived from a "Concept Dependency Graph" of skills). This allows for more granular curriculum construction.

3. Dataset Insights: A byproduct of the method is the potential to create a "cartography" of a dataset, offering insights into skill coverage and relationships between concepts.

Weaknesses:

1. Reliance on a Capable Teacher Model: The entire framework heavily depends on the availability and proficiency of a strong "teacher" LLM (like GPT-40) that can accurately reason, decompose problems, and generate correct sub-tasks. The quality of the generated curriculum is tied to the teacher's capabilities.

2. Incomplete Evaluation Scope: The paper primarily evaluates the method's effectiveness on base student models (Qwen2.5-1.5B and Qwen3-4B-Base). It does not include experiments applying the DECOMPX training methodology directly to reasoning models of comparable sizes to see if similar or enhanced gains are achieved on reasoning models.

3. Limited Comparison with State-of-the-Art Distillation: This approach shares some similarities with knowledge distillation, as both leverage capability transfer from larger models to enhance smaller ones. However, experimental results demonstrate that neither the 4B nor 1.5B parameter models can match the performance of the DeepSeek-R1-Distill-Qwen-1.5B model, which is directly distilled from Deepseek-R1.

---

> ### Author Rebuttal · Authors · 2025-07-31
>
> We thank the Reviewer for the insightful questions and comments. We appreciate the recognition of the novelty of our decomposition-based curriculum learning approach, the principled difficulty scoring mechanism, and the useful insights enabled by our dataset analysis framework. We address each question and concern below:
>
> **Q1 & W3**
>
> > What advantages does it offer over directly distilling with a reasoning model?
>
> We appreciate the opportunity to clarify the advantages of our method compared to direct distillation.
>
> First, given identical original data (same prior knowledge) and the same teacher model, our proposed dataset decomposition approach outperforms direct distillation. Furthermore, our approach is compatible with existing distillation methods, enabling further performance improvements.
>
> We conducted a baseline experiment of direct distillation from the same teacher model (GPT-4o) and the same original data (MATH training subset and AIME 2024, respectively). The comparison results shown in the table demonstrate the advantages of our method over direct distillation.
>
> | Model (based on Qwen2.5-1.5B) | MATH 500 |
> | --- | --- |
> | SFT-Direct Distillation | 49.6 ± 2.2 |
> | SFT-DecompX (Ours) | 50.8 ± 2.2 |
> | SFT-DecompX + Curriculum (Ours) | 51.6 ± 2.2 |
>
> | Task (based on Qwen3-4B-Base) | AIME 2025 |
> | --- | --- |
> | SFT-Direct Distillation | 6.67 ± 4.6 |
> | SFT-DecompX (Ours) | 13.3 ± 6.3 |
> | SFT-DecompX + Curriculum (Ours) | 16.7 ± 6.9 |
>
> **Q2 & W2**
>
> > However, experimental results demonstrate that neither the 4B nor 1.5B parameter models can match the performance of the DeepSeek-R1-Distill-Qwen-1.5B model, which is directly distilled from Deepseek-R1.
>
> We respectfully point out that the comparison is not fair for three reasons. **(1) Training data scale and prior knowledge.** DeepSeek-R1-Distill-Qwen-1.5B is trained on ~800k SFT samples curated with DeepSeek-R1, which injects substantially more prior knowledge than our setup. In contrast, our training data are purely derived from a small MATH training subset and AIME 2024 via decomposition, without access to large external corpora. **(2) Different base models.** The base for DeepSeek-R1-Distill-Qwen-1.5B is Qwen2.5-Math-1.5B (math-specialized), whereas our 1.5B and 4B students are base models (e.g., Qwen2.5-1.5B Base, Qwen3-4B Base) without math specialization; this materially advantages the distilled model. **(3) Compute and reproducibility.** The distilled training set for DeepSeek-R1-Distill-Qwen-1.5B is not publicly released, making it impossible to match compute/data budgets or to run controlled re-trainings under identical conditions.
>
> Crucially, our goal is different and complementary: we show a cost-effective pathway to improve small models’ reasoning by structuring limited supervision. Training R1-distilled models at the 800k-example scale is not inexpensive; our method offers orders-of-magnitude lower data/compute while achieving strong gains and better generalization from easy-to-hard progression.
>
> Finally, our approach is compatible with direct distillation rather than a replacement. Decomposition can augment distillation by (1) filtering/structuring teacher signals into prerequisite-aligned sub-tasks, and (2) staging training by difficulty to stabilize optimization for small models. We view combining R1-style distillation with our decomposition-driven curriculum as a promising direction and are actively exploring it.
>
> > Is this method effective for reasoning models? Can it further enhance their performance?
>
> Thank you for your question. Given time constraints and the significant cost of evaluating reasoning models, we are currently actively conducting experiments on reasoning models, to evaluate the effectiveness of our method. We will update our results and provide full metrics as soon as the experiments are complete.
>
> **Q3 & W1**
>
> > How dependent is this method on the capability of the Teacher Model? Can it still achieve good results when the Teacher Model itself performs poorly?
>
> We appreciate this important question and would like to clarify both the scope and contributions of our work.
>
> Our method is designed to leverage the compositionality of language reasoning to enhance small models’ learning, especially in challenging reasoning tasks. A key motivation for our approach is the observation that small models often struggle to learn complex reasoning patterns from randomly shuffled data. Our recursive dataset decomposition explicitly addresses this by breaking down complex problems into hierarchical sub-problems, enabling a more controlled and interpretable learning curriculum.
>
> As stated in our paper’s limitations (Section 5), we assume access to a reasonably capable teacher model to generate valid decompositions. This assumption is shared with many prior works in knowledge distillation. Importantly, our method includes built-in verification steps and retry mechanisms to ensure the correctness of generated subproblems, mitigating the risk of low-quality outputs from imperfect teacher models.
>
> While the method’s current implementation relies on a relatively strong teacher (e.g., GPT-4o), we view exploring the weak-to-strong generalization regime, where smaller or less capable models can still serve as effective teachers for equally-sized or even larger students, as an exciting direction for future work. We acknowledge this as a meaningful challenge and intend to explore it in subsequent research.
>
> We also emphasize that our approach offers distinct and non-replaceable advantages over traditional distillation methods. Specifically, it enables (1) granular control over the learning trajectory of small models, (2) interpretability via explicit sub-task structuring, and (3) curriculum-guided training, which is especially critical for reasoning tasks where intermediate skills are essential.
>
> In practice, small base models are highly desirable due to their lower cost and hardware efficiency, and are increasingly deployed in real-world scenarios. Our framework provides a principled way to improve their reasoning capabilities via structured and concept-aware training data—an area where direct distillation alone often lacks fine-grained control or gradual guidance.
>
> In summary, while our method benefits from a strong teacher model, its design offers robustness and generalizability, and opens up promising future directions for more flexible teacher-student configurations.
>
> ---
>
> We thank the reviewer again for their constructive comments. We would be more than happy to further discuss if there are any remaining concerns or questions!

---

> > ### Comment · Reviewer_nBar · 2025-08-04
> > **Good Response**
> >
> > Thank you for the detailed rebuttal and clarification. I appreciate the additional experiments, which provide some evidence that the proposed method can outperform direct distillation under certain small-scale settings. Based on these clarifications, I have increased my score.
> >
> > That said, I still have some reservations. The generalization ability of the method remains unclear, especially in out-of-distribution scenarios (e.g., training on a subset of MATH and testing on unseen datasets like AIME 2025). Furthermore, while the method appears promising, it has not yet demonstrated clear advantages over direct distillation from strong reasoning models, particularly on larger benchmarks or with publicly available competitive baselines such as DeepSeek-R1-Distill. Finally, its effectiveness in enhancing reasoning-capable student models remains to be more convincingly demonstrated.
> >
> > Overall, the paper takes a step in an interesting direction, but key questions regarding generality and relative benefits remain open.

---

> ### Author Response · Authors · 2025-08-05
>
> We sincerely thank the reviewer for acknowledging our detailed rebuttal and for increasing the score based on the additional clarifications. We address your remaining reservations as follows:
>
>
> > **The generalization ability of the method remains unclear, especially in out-of-distribution scenarios (e.g., training on a subset of MATH and testing on unseen datasets like AIME 2025).**
>
> We conducted the exact experiment suggested -- training on MATH subset and evaluating on the more challenging AIME 2025 dataset, and we observed strong improvements:
>
> | Model                                                | AIME 2025        |
> | ---------------------------------------------------- | ---------------- |
> | DeepSeek-R1-Distill-Qwen-1.5B                        | 23.33 ± 7.85     |
> | DeepSeek-R1-Distill-Qwen-1.5B-DecompX (Ours) | **30.00 ± 8.51** |
>
> These results directly confirm that our proposed method improves generalization from MATH to AIME2025, even under a substantial distribution shift in both difficulty and style.
>
> Beyond this setting, we have also systematically evaluated OOD generalization in other scenarios:
>
> 1. **Same-domain, cross-year generalization**: Training on decomposed AIME 2024 and testing on AIME 2025, as shown in Table 2 of our paper, with promising results positively recognized by Reviewer A4Mp, who explicitly noted that "Models trained with DECOMPX exhibit better generalization, particularly on the AIME2025 dataset which represents an OOD shift from AIME2024."
>
> 2. **Cross-language & cross-subdomain generalization in code generation**: To further demonstrate the generalization capabilities and generality of our method, we conducted additional experiments in a non-mathematical domain that requires complex reasoning: coding, using the CodeForces-CoTs competitive programming dataset (with solutions in C++) for training and the HumanEval benchmark (Python function completion) for evaluation, which is an explicitly out-of-distribution scenario. Our results, using Qwen2.5-1.5B-Instruct and DeepSeek-R1-Distill-Qwen-1.5B as the base models, show substantial improvements:
>
> | Model | HumanEval pass@1 |
> | --- | --- |
> | Qwen2.5-1.5B-Instruct | 34.15 ± 3.71 |
> | Qwen2.5-1.5B-Instruct-SFT | 35.37 ± 3.74 |
> | Qwen2.5-1.5B-Instruct-DecompX (Ours) | **42.68 ± 3.87** |
> | DeepSeek-R1-Distill-Qwen-1.5B | 36.01 ± 1.85 |
> | DeepSeek-R1-Distill-Qwen-1.5B-DecompX (Ours) | **57.90 ± 0.98** |
>
> In summary, this broad set of results (from MATH→AIME2025, to AIME2024→AIME2025, to CodeForces→HumanEval) demonstrates that DecompX's generalization benefits are consistent across tasks, domains, and types of distribution shifts.
>
> > **While the method appears promising, it has not yet demonstrated clear advantages over direct distillation from strong reasoning models, particularly on larger benchmarks or with publicly available competitive baselines such as DeepSeek-R1-Distill.**
>
> We directly addressed this concern by comparing our method against the publicly available and competitive baseline DeepSeek-R1-Distill following the experimental setup above (with o4-mini as the teacher model). Our results in the table below (*Qwen2.5-1.5B-Instruct-DecompX (Ours) vs DeepSeek-R1-Distill-Qwen-1.5B*) show that our proposed method offers advantages over the competitive DeepSeek-R1-Distill model of the same size.
>
> > **Its effectiveness in enhancing reasoning-capable student models remains to be more convincingly demonstrated.**
>
> We also fine-tuned the DeepSeek-R1-Distill-Qwen-1.5B model (reasoning model as the student model) with our decomposed data, and observed a dramatic improvement, clearly demonstrating our method’s significant effectiveness in enhancing reasoning-capable student models. The following results (*DeepSeek-R1-Distill-Qwen-1.5B-DecompX (Ours) vs DeepSeek-R1-Distill-Qwen-1.5B*) underscore this advantage on different reasoing domains:
>
> | Model                                             | AIME 2025        |
> | ---------------------------------------------------- | ---------------- |
> | DeepSeek-R1-Distill-Qwen-1.5B                        | 23.33 ± 7.85     |
> | DeepSeek-R1-Distill-Qwen-1.5B-DecompX (Ours) | **30.00 ± 8.51** |
>
> | Model | HumanEval pass@1 |
> | --- | --- |
> | Qwen2.5-1.5B-Instruct | 34.15 ± 3.71 |
> | Qwen2.5-1.5B-Instruct-SFT | 35.37 ± 3.74 |
> | Qwen2.5-1.5B-Instruct-DecompX (Ours) | **42.68 ± 3.87** |
> | DeepSeek-R1-Distill-Qwen-1.5B | 36.01 ± 1.85 |
> | DeepSeek-R1-Distill-Qwen-1.5B-DecompX (Ours) | **57.90 ± 0.98** |
>
> ---
> We appreciate the reviewer’s thoughtful feedback and believe these additional clarifications further demonstrate the strengths and generality of our method. We would be more than happy to further discuss if there are any remaining concerns or questions!

---

> ### Author Response · Authors · 2025-08-09
> **Looking forward to continuing our discussion**
>
> Thank you again for your recognition of our work and thoughtful rebuttal feedback! We truly appreciate your time and constructive suggestions. We're pleased that our response has helped address your concerns. Following your suggestions, we have added the MATH→AIME2025 experiment (demonstrating clear out-of-distribution gains), clarified the existing OOD experiments in our paper, and included comprehensive comparisons with DeepSeek-R1-Distill. These updates are now reflected in our rebuttal and comment above.
>
> At its core, our work demonstrates how to leverage existing models as an efficient and effective generator of high-quality synthetic curricula that teach small or new models complex reasoning patterns. The key idea is to create a high-quality synthetic curriculum that teaches complex concepts by shaping data for learning with gradual difficulty levels, rather than simply augmenting the dataset. This can also establish a foundation for recursive model self-improvement, where each generation of models enhances both the quality and curriculum structure of training data for the new version of models using test-time compute -- We leave this as a promising future direction, especially at scale, where our proposed hierarchical data decomposition and curriculum learning could have even greater impact.
>
> **If there are any remaining questions or concerns, we would be very happy to discuss and address them within the remaining discussion period!**

---

### Official Review · Reviewer_UED6 · 2025-06-29

**Clarity:** 3
**Significance:** 2
**Originality:** 3
**Rating:** 4
**Confidence:** 2

**Summary:**

The authors propose a new method for generating curriculum (for curriculum learning), in which the teacher model recursively breaks down task into increasingly simple components for the student model to learn the curriculum in increasing difficulty. A new scoring system is proposed for measuring data difficulty during curriculum construction. Experiments are conducted on math datasets.

**Questions:**

Besides the issues noted weaknesses section, it would also be good to see if this method offers improvement on non-math datasets. The results in this submission currently run a bit thin.

Could you also explain how you implemented Line 8 of Algorithm 1 in your experiments (i.e., what auto-rater you used and how consistency is checked)?

An interesting but not necessary thing would be to see if the proposed method can still offer big improvement when the teacher and student models have similar size, which could show some promise of self-improvement.

**Ethical Concerns:**

["NO or VERY MINOR ethics concerns only"]

**Final Justification:**

The rebuttal from the authors have cleared some of my previous questions on some choices made in the experiment setup.

**Limitations:**

Limitations have been sufficiently discussed in the conclusion section.

**Quality:**

3

**Strengths And Weaknesses:**

Strengths:
The writing is clear. The proposed method is convincing. I appreciate the case study in the submission.

Weaknesses:
I don’t think the experiment results are strong enough to demonstrate the superiority of the proposed method yet. First of all, only one other method is compared with in the experiments. More importantly, while I understand the reason behind some of the current choices of # data in Table 1 and 2, it would make more sense to have this number leveled across DecompX and comparison experiments. For example, both DecompX and DecompX+Curriculum use 4500 in Table 1 while MetaMATH-Aug only uses 2638 and SFT only 360. This makes the higher score of DecompX less convincing.

---

> ### Author Rebuttal · Authors · 2025-07-31
>
> Thank you for your valuable feedback and constructive suggestions. We appreciate your recognition of our clear writing, convincing methodology, and the illustrative case study provided.
>
> **W1**
>
> > Only one other method is compared with in the experiments.
>
> We conducted a baseline experiment of direct distillation from the same teacher model (GPT-4o) and the same original data (MATH training subset and AIME 2024, respectively). The comparison results shown in the table demonstrate the advantages of our method over direct distillation.
>
> | Model (based on Qwen2.5-1.5B) | MATH 500 |
> | --- | --- |
> | SFT-Direct Distillation | 49.6 ± 2.2 |
> | SFT-MuggleMath | 50.4 ± 2.2 |
> | SFT-DecompX (Ours) | 50.8 ± 2.2 |
> | SFT-DecompX + Curriculum (Ours) | 51.6 ± 2.2 |
>
> | Task (based on Qwen3-4B-Base) | AIME 2025 |
> | --- | --- |
> | SFT-Direct Distillation | 6.67 ± 4.6 |
> | SFT-MuggleMath | 6.67 ± 2.2 |
> | SFT-DecompX (Ours) | 13.3 ± 6.3 |
> | SFT-DecompX + Curriculum (Ours) | 16.7 ± 6.9 |
>
> To further address this concern, we conducted an additional baseline using the MuggleMath dataset (147,787 examples). Evaluated on the same benchmarks, this baseline underperforms our approach despite leveraging substantially more training data and richer prior knowledge from the seed datasets (MATH and GSM8K). These results further support the advantage of our structured, decomposition-based curriculum.
>
> **W2**
>
> > It would make more sense to have this number leveled across DecompX and comparison experiments.
>
>
> The discrepancy in the number of data points used between DecompX and other methods (e.g., MetaMATH-Aug and SFT) arises naturally due to our recursive decomposition process, which inherently generates multiple subproblems from each original problem. Importantly, our experiments were structured to provide a fixed knowledge budget. Thus, even though we used more decomposed samples, each smaller dataset (e.g., MetaMATH-Aug and SFT) received proportionally increased training epochs to maintain a fair comparison. This design choice demonstrates our method's effectiveness in enhancing performance without introducing new knowledge or data.
>
> To ensure a fair comparison, all methods start from the same original data (the 30 AIME2024 problems), serving as prior knowledge. We control the total number of training tokens by adjusting training epochs inversely proportional to the ratio of token counts between different methods. This ensures that all approaches receive an equal compute budget. Therefore, the observed performance gains of our method are not due to more data or compute, but rather the improved structure and utility of the augmented data. The improved generalization to AIME2025 underscores the effectiveness and robustness of our curriculum design.
>
> **Q1: Generalization to Non-Mathematical Datasets**
>
> Thank you for raising this valuable point. We have indeed explored our method's effectiveness beyond mathematics, applying it to a coding domain using the competitive programming dataset Codeforces-CoTs and the HumanEval benchmark. The base model was Qwen2.5-1.5B-Instruct. Our method resulted in significant performance improvements:
>
> | Model | HumanEval pass@1 |
> | --- | --- |
> | Qwen2.5-1.5B-Instruct | 34.15 ± 3.71 |
> | Qwen2.5-1.5B-Instruct-SFT | 35.37 ± 3.74 |
> | Qwen2.5-1.5B-Instruct-DecompX (Ours) | 42.68 ± 3.87 |
>
> These results clearly demonstrate our method’s potential in non-mathematical domains.
>
> **Q2: Implementation of Line 8 (Algorithm 1)**
>
> > Could you also explain how you implemented Line 8 of Algorithm 1 in your experiments (i.e., what auto-rater you used and how consistency is checked)?
>
> We clarify that the step employs an auto-solver (rather than an auto-rater). This refers to the process of generating a standalone answer to the subproblem and checking its consistency, as explained in Lines 112–117 of the main paper. Specifically, for each generated subproblem, the teacher model solves it twice: Once with access to the original context from which the step is derived; Once without any context, relying solely on the generated question. We then compare the two answers. Only when the answers match (verified by `Math-Verify`) do we accept the subproblem. Otherwise, we retry the process with small rephrasings until consistency is achieved or a retry budget is exhausted (Those that fail verification will be filtered out). It provides a reasonable and scalable mechanism to ensure basic correctness and filter out obviously invalid generations.This step is crucial to ensure that the generated subproblem is genuinely solvable and independent, thereby maintaining the reliability and trustworthiness of the decomposed dataset.
>
> **Q3: Teacher-Student Model Size Similarity:**
>
> This is indeed an insightful suggestion. Our experiments indicate that smaller models struggle to produce clean and structured decompositions. Exploring the "weak-to-strong" generalization, where smaller models help train similarly-sized or even larger models, represents an exciting avenue for future research. We acknowledge this challenge and intend to pursue it in subsequent studies.
>
> ---
>
> We thank the reviewer again for their constructive comments. We would be more than happy to further discuss if there are any remaining concerns or questions!

---

> > ### Comment · Reviewer_UED6 · 2025-08-02
> >
> > I appreciate the clarification from the authors and have raised my score accordingly.

---

> ### Author Response · Authors · 2025-08-09
>
> We are grateful for the reviewer's engagement and for confirming that our response resolved your concerns!
>
> ---
>
> We would like to follow up on our previous response to the inspiring question **Q3** about the potential of our method for self-improvement when teacher and student models have comparable capacity, and add an additional perspective:
>
> We thank the reviewer again for this suggestion, and we agree that such experiments would provide stronger evidence for self-improvement. While our current work adopts a “large teacher → small student” setup, the method was designed to leverage the previous generation of models not only as data consumers but also as data generators. The key idea is to create a high-quality synthetic curriculum that teaches complex concepts by shaping data for learning, rather than simply augmenting the dataset.
>
> This design naturally supports scenarios where teacher and student have similar capacity: the benefit comes not from the size gap but from the teacher’s ability to organize and distill knowledge into teaching-oriented examples. In the weak-to-strong generalization setting, this can be achieved as long as the weaker model is already capable of producing clean and well-structured decompositions. We leave this as a promising future direction, especially at scale, where our proposed hierarchical data decomposition and curriculum learning could have even greater impact.

---

### Official Review · Reviewer_eiT7 · 2025-07-01

**Clarity:** 4
**Significance:** 3
**Originality:** 3
**Rating:** 5
**Confidence:** 2

**Summary:**

This paper proposes a curriculum learning approach built upon the concept of distillation/data augmentation. Applied to math problem solving, the approach entails decomposing math problems into subproblems based on reasoning steps generated by LLM. These subproblems are generated recursively and organized into a tree structure, tagged with difficulty levels using structural and conceptual features. Empirical results show marginal improvements over alternative methods on the training set but demonstrate significant gains on unseen datasets (AIME 2025) with encouraging generalization promise.

**Questions:**

1. Have you analyzed failure cases, particularly on challenging problems like AIME, to better understand the limitations of your approach?

2. Have you considered human evaluation to validate the generated tags and difficulty levels, ensuring they align with human intuition?

3. How well do you expect your method to generalize beyond procedural domains like math? do you have insights on what types of tasks is it best suited for and what are not suitable for?

**Ethical Concerns:**

["NO or VERY MINOR ethics concerns only"]

**Limitations:**

The limitation was lightly discussed. I wish the authors could also discuss the social impact of this work. While it is interesting to design AI to be as smart as possible, the authors could also discuss how this approach can be used to improve math education, for example, using the approach to automatically generate scaffolding, determine the difficulty level, and tag the problem domains are very interesting contributions to math education and beyond.

**Paper Formatting Concerns:**

no concern.

**Quality:**

3

**Strengths And Weaknesses:**

Quality: This paper is well written, and the presentation of the proposed methods is both clear and sound, supported by solid experimental results. The work would be further strengthened by a deeper analysis of the results—for example, by examining failure cases to better understand the limitations of the approach. Additionally, incorporating human validation of the generated tags and difficulty levels would further enhance the credibility of the decomposition approach.

Clarity: Overall, the authors have done a decent job in presenting their ideas. The paper is clearly written, and the inclusion of illustrative examples and detailed algorithm descriptions supports comprehension. The use of a case study to demonstrate the decomposition process is particularly appreciated, as it offers concrete insight into how the proposed method operates in practice.

Significance: The paper presents a knowledge distillation approach to curriculum learning, demonstrated through the task of mathematical problem solving. While the authors suggest that the method can be generalized, a more detailed discussion on its broader applicability would be valuable. For instance, the approach appears well-suited for procedure-based problem solving, but its effectiveness in other domains remains unclear. Additionally, the definition of difficulty in the paper seems primarily tied to the number of steps required to complete a task. However, in some cases, difficulty also involves the need for non-obvious insights—such as re-framing a problem to allow for an elegant, minimal solution. This may help explain why, despite significant efforts (including the proposed enhancements), performance on solving new AIME problems remains relatively low.

Originality: The paper is situated in the literature of data augmentation/distillation with LLM as well as curriculum learning. The related work is detailed and relevant. The paper mentioned that the idea of curriculum learning with synthetic data is not entirely novel. I wish the authors could articulate better the contribution of this paper in the context of other similar methods, such as Tiny Stories, understanding it is from different domains.

---

> ### Author Rebuttal · Authors · 2025-07-31
>
> We thank the reviewer for their positive and thoughtful feedback and address their specific questions and suggestions below:
>
> **Q1**
>
> > Have you analyzed failure cases, particularly on challenging problems like AIME, to better understand the limitations of your approach?
>
> As stated in our paper’s limitations (Section 5), the limitations of our approach are inherently bounded by the capabilities of the teacher model. Specifically, when the teacher model fails to conduct the Step Extraction in Section 3.1, the whole decomposition process is less effective.
>
> However, we would also like to point to some successful cases in Appendix 5 that illustrate how our method effectively deals with problems requiring "non-obvious insights". For instance, in the whole-prime-power assignment scenario (ab=N with gcd(a,b)=1), the key leap is recognizing that each prime power must be wholly assigned to either a or b; our Concept Dependency Graph captures this high conceptual depth (linking Fundamental Theorem of Arithmetic → coprimality → binary counting), so the curriculum surfaces the insight early. Similarly, for non-square symmetry halving (counting ordered factor pairs (a,b) of non-square N), the crucial step is seeing that a fixed point (a=a) occurs only for perfect squares; by tying parity tests to symmetry arguments, our graph again flags the step as concept-heavy despite its short reasoning chain, ensuring the curriculum allocates it appropriate emphasis.
>
> **Q2**
>
> > Have you considered human evaluation to validate the generated tags and difficulty levels, ensuring they align with human intuition?
>
> Yes. We appreciate the reviewer’s suggestion regarding human evaluation. Our manuscript includes illustrative examples in Table 3, Appendix 2, and Appendix 5, showing strong alignment of our automatically generated tags and difficulty levels with human intuition. Additionally, Appendix 3 provides further validation through zero-shot performance analysis of the Qwen3-4B-Base model across difficulty tiers.
>
> We would like to highlight that, given our method's generic and automated nature, it uniquely demonstrates broad applicability without relying on domain-specific prior knowledge or human expert intervention. Thus, our approach significantly reduces human cost by enabling effective generation of simpler problems independently of domain experts.
>
> **Q3**
>
> > How well do you expect your method to generalize beyond procedural domains like math? do you have insights on what types of tasks is it best suited for and what are not suitable for?
>
> To explore generalization beyond purely procedural domains, we conducted additional experiments in the coding domain. We used the competitive programming dataset CodeForces-CoTs for training and HumanEval as our evaluation benchmark, and Qwen2.5-1.5B-Instruct as the base model. Our results demonstrate significant performance improvements using our approach:
>
> | Model | HumanEval pass@1 |
> | --- | --- |
> | Qwen2.5-1.5B-Instruct | 34.15 ± 3.71 |
> | Qwen2.5-1.5B-Instruct-SFT | 35.37 ± 3.74 |
> | Qwen2.5-1.5B-Instruct-DecompX (Ours) | 42.68 ± 3.87 |
>
> These results underline the broader applicability of our decomposition method in domains that require structured reasoning, such as programming and symbolic reasoning. Conversely, less structured tasks or tasks demanding high levels of creativity or subjective interpretation (e.g., narrative storytelling or literary analysis) might benefit less from our structured curriculum approach.
>
> **Weakness**
> > I wish the authors could articulate better the contribution of this paper in the context of other similar methods, such as Tiny Stories, understanding it is from different domains.
>
> TinyStories highlights that data quality often outweighs data quantity for small language models. However, unlike our approach, TinyStories does not incorporate difficulty-based curriculum construction. Moreover, our method provides a generic synthetic data generation framework, contrasting with the dataset-specific focus of TinyStories.
>
> **Limitation: Social Impact and Educational Potential**
>
> Following the reviewer’s insightful suggestion, we will further emphasize the educational potential and social impact of our method, particularly in math education. Our approach can facilitate automatic scaffolding, dynamically build knowledge graphs, and design better curricula by precise difficulty assessments, thereby enhancing personalized learning experiences and supporting equitable educational practices. We will add this discussion to the revised manuscript.
>
> ---
>
> We thank the reviewer again for their constructive comments. We would be more than happy to further discuss if there are any remaining concerns or questions!

---

> > ### Comment · Reviewer_eiT7 · 2025-08-04
> >
> > appreciate the responses. I keep the same scores.

---

> > > ### Author Response · Authors · 2025-08-09
> > >
> > > Thank you for your positive feedback and for maintaining your recommendation. Your thorough review has been very valuable! We will integrate your suggestions to enrich our paper.

---

### Official Review · Reviewer_A4Mp · 2025-07-01

**Clarity:** 3
**Significance:** 3
**Originality:** 3
**Rating:** 4
**Confidence:** 4

**Summary:**

This paper introduces DECOMPX, a novel curriculum learning approach that recursively decomposes complex problems into simpler, more learnable components.
The authors propose a teacher-student framework where a powerful teacher model (GPT-4o) generates easier versions of examples by reasoning step-by-step.
The method also includes a novel scoring system for data difficulty based on structural complexity and conceptual depth, enabling curriculum construction over the decomposed data.
Experiments on MATH and AIME datasets demonstrate improved performance compared to standard training baselines.

**Questions:**

* Could the authors provide more details on the specific prompts used for the teacher model during "Step Extraction" and "Concept Tagging"? How is the initial set of concept tags defined or discovered, and is there any pre-defined ontology or does it emerge purely from the teacher model's responses?
* The description of training sets is somewhat vague. Could the authors clarify if Table 1 and Table 2 both utilize AIME2024 as the training set for their respective models, or if Table 1 uses the MATH dataset's training portion? Why DECOMPX use less data than SFT baseline in Table1 but more data in Table2. A detailed introduction of the training data for each experiment would be beneficial.
* How is it guaranteed that every extracted "step" will always have a well-defined, verifiable answer? Some intermediate steps in a human-like reasoning chain might involve logical inferences, variable assignments, or reflections that do not directly yield a answer that can be "solved" or "verified" independently as a sub-problem. How does the current verification mechanism handle such cases, or are these types of steps implicitly ignored?

**Ethical Concerns:**

["NO or VERY MINOR ethics concerns only"]

**Final Justification:**

Most of issues are resolved, including the clarification of long CoTs, the relative weak baseline, the detailed definition of "step", and other details. I choose to increase my score to 4.

**Limitations:**

yes

**Quality:**

3

**Strengths And Weaknesses:**

Strengths:
* The core idea of recursively decomposing complex problems into simpler sub-problems, coupled with extracting and organizing "concept tags" into a dependency graph, is novel.
* Models trained with DECOMPX exhibit better generalization, particularly on the AIME2025 dataset which represents an out-of-domain distribution shift from AIME2024.
* The proposed scoring system, combining structural complexity and conceptual depth, provides a principled way to quantify problem difficulty within the decomposed dataset.

Weaknesses:
* The experiments appear to primarily focus on the decomposition of relatively short CoTs, lacking an in-depth exploration of how the recursive decomposition method performs and scales with long CoT solutions, which are common in complex mathematical reasoning problems and large reasoning models like DeepSeek-R1 and Qwen3.
* While the paper highlights improvements on AIME, the AIME2024 training set used consists of only 30 problems. This is an extremely small dataset for fine-tuning language models in mathematical reasoning, making the observed performance gains potentially sensitive to this limited data size. The generalizability of conclusions drawn from such a small training set to broader mathematical reasoning tasks remains a concern.
* The comparison with MetaMath-Aug, particularly with only 114 data points for AIME, appears to be a relatively weak baseline. The paper could benefit from comparing against a wider range of more recent and robust data augmentation methods in mathematical reasoning.

---

> ### Author Rebuttal · Authors · 2025-07-31
>
> We thank the reviewer for the detailed review and questions. We appreciate the recognition of the novelty of our recursive decomposition framework, the effectiveness of the concept tag dependency graph, the principled difficulty scoring system, and the improved generalization performance, especially under the OOD setting.
>
> Below, we address each concern and question raised:
>
> **W1**
>
> > The experiments appear to primarily focus on the decomposition of relatively short CoTs, lacking an in-depth exploration of how the recursive decomposition method performs and scales with long CoT solutions, which are common in complex mathematical reasoning problems and large reasoning models like DeepSeek-R1 and Qwen3.
>
> Thank you for raising this point. We interpret “long CoTs” in two ways and address both:
>
> * Long gold solutions for the original problems: Our method assumes access to a high-quality reference solution, which is standard in mathematical-reasoning benchmarks. On datasets such as AIME - representative in both problem complexity and solution length - our recursive decomposition performs reliably.
>
> * Long, model-generated traces (e.g., DeepSeek-R1, Qwen3): The framework is model-agnostic. We can control the length of each generated token number.
>
> Our experimental setup follows prior work and targets benchmarks commonly used. We agree that pushing to even longer-horizon tasks, e.g., automated theorem proving or high-dimensional geometry, would be valuable. While beyond our current scope, we view this as a natural extension and are happy to incorporate specific cases (e.g., very long traces from R1/Qwen3) if suggested. If this does not fully capture your concern, we welcome clarifications or pointers to target datasets/trace lengths you would like us to evaluate.
>
> **W2**
> > While the paper highlights improvements on AIME, the AIME2024 training set used consists of only 30 problems. This is an extremely small dataset for fine-tuning language models in mathematical reasoning, making the observed performance gains potentially sensitive to this limited data size. The generalizability of conclusions drawn from such a small training set to broader mathematical reasoning tasks remains a concern.
>
> We would like to clarify that our goal is to demonstrate how a more effective data curriculum enabled by recursive decomposition can amplify a limited supervision signal for a complex problem into a more comprehensive and gradual training set.
>
> To ensure a fair comparison, all methods start from the same original data (the 30 AIME2024 problems), serving as prior knowledge. We control the total number of training tokens  by adjusting training epochs inversely proportional to the ratio of token counts between different methods. This ensures that all approaches receive an equal compute budget. Therefore, the observed performance gains of our method are not due to more data or compute, but rather the improved structure and utility of the augmented data. The improved generalization to AIME2025 underscores the effectiveness and robustness of our curriculum design.
>
> **W3:**
>
> > The comparison with MetaMath-Aug appears to be a relatively weak baseline. The paper could benefit from comparing against a wider range of more recent and robust data augmentation methods in mathematical reasoning.
>
> We conducted a baseline experiment of direct distillation from the same teacher model (GPT-4o) and the same original data (MATH training subset and AIME 2024, respectively). The comparison results shown in the table demonstrate the advantages of our method over direct distillation.
>
> To further address this concern, we conducted an additional baseline using the MuggleMath dataset (147,787 examples). Evaluated on the same benchmarks, this baseline underperforms our approach despite leveraging substantially more training data and richer prior knowledge from the seed datasets (MATH and GSM8K). These results further support the advantage of our structured, decomposition-based curriculum.
>
>
> | Model (based on Qwen2.5-1.5B) | MATH 500 |
> | --- | --- |
> | SFT-Direct Distillation | 49.6 ± 2.2 |
> | SFT-MuggleMath | 50.4 ± 2.2 |
> | SFT-DecompX (Ours) | 50.8 ± 2.2 |
> | SFT-DecompX + Curriculum (Ours) | 51.6 ± 2.2 |
>
> | Task (based on Qwen3-4B-Base) | AIME 2025 |
> | --- | --- |
> | SFT-Direct Distillation | 6.67 ± 4.6 |
> | SFT-MuggleMath | 6.67 ± 2.2 |
> | SFT-DecompX (Ours) | 13.3 ± 6.3 |
> | SFT-DecompX + Curriculum (Ours) | 16.7 ± 6.9 |
>
>
> **Q1**
> > How is the initial set of concept tags defined or discovered, and is there any pre-defined ontology or does it emerge purely from the teacher model's responses?
>
> To maintain the general applicability of our method, we deliberately avoided pre-defined ontologies; all concept tags are generated dynamically from the teacher model’s responses.
>
> > Could the authors provide more details on the specific prompts used for the teacher model during "Step Extraction" and "Concept Tagging"?
>
> - **Step Extraction Prompt:**
>
> ```Python
> prompt = f"""
> Split the following solution into at most {max_steps} reasoning steps.
> Ensure the steps are balanced in length and complexity.
> Only split when the step introduces a new **mathematical operation**.
> If the step is already atomic, do not split further.
>
> Format your response as:
> Step 1: [first step content]
> Step 2: [second step content]
> ...and so on up to at most Step {max_steps}.
> Do not include any additional text or explanation.
>
> Solution:
> {solution}
> """
> ```
> - **Concept Tagging Prompt:**
>
> ```Python
> tag_prompt = f"""
> Given the following reasoning step from solving a math problem:
>
> \"{step}\"
>
> Output only one precise, specific high-school level mathematical concept (tag) relevant to this step.
> This concept should represent the core mathematical knowledge required to understand and perform this step.
>
> Important requirements:
> 1. DO NOT use broad categories like "Algebra", "Geometry", or "Number Theory".
> 2. Identify the exact skill or technique being used (e.g., "Pythagorean Theorem", "Quadratic Formula", "GCD Calculation").
> 3. The concept should be atomic and specific enough to fit into a skill dependency graph.
> """
> ```
>
> **Q2: Clarification on Training Data (Tables 1 & 2):**
>
> For Table 1, the baseline `SFT (full dataset)` refers to the original MATH dataset training portion (7500 examples), as explicitly mentioned in our paper (Line 215). Appendix 1 further details our balanced sampling subset for another baseline `SFT`. Our baseline `SFT-MetaMATH-Aug` and `DecompX` methods are generated from this subset (360 examples), chosen due to the extensive computational requirements of decomposing each problem. In Table 2, the AIME2024 dataset (initially 30 problems) was expanded into 385 subproblems using our decomposition method, surpassing the number of data points in the SFT baseline.
>
> **Q3: Ensuring Verifiable Answers for Extracted Steps:**
>
> Subproblems and answers are not directly generated from random steps; instead, they are generated based on concept tags that ensure each subproblem is associated with a specific mathematical operation from the step extraction. In the Appendix 4 of our submission, we listed the tag clusters identified in both the MATH and the AIME dataset for your reference, where there are no tags like `logical inferences`, `variable assignments`, or `reflections`.
>
> The subproblem generation prompt explicitly requires questions to yield a "specific numerical answer," ensuring solvability and verifiability:
>
> ```Python
> combined_prompt = f"""
> You are helping to decompose and solve a math problem.
>
> Original problem:
> \"{problem}\"
>
> Given the following reasoning step from solving the original problem:
> \"{step}\"
>
> and the relevant mathematical concept:
> \"{tag_topic}\"
>
> Your tasks:
> 1. Generate a self-contained and standalone high school-level math **question** that:
>     - Is grounded in the **context of the original problem**.
>     - Asks specifically ONLY about the mathematical operation or concept in this individual step: {tag_topic}
>     - Does NOT include or reference information from earlier or later reasoning steps
>     - Requires a **specific numerical answer**.
> 2. Solve the question with a detailed step-by-step **explanation** based on this reasoning step and original context.
> 3. Output only a final **numerical answer**.
>
> Strict Format:
> Question: [your generated math question]
> Explanation: [step-by-step reasoning]
> |||
> Answer: [final number only]
> """
> ```
>
>
> A verification process further ensures the correctness of generated subproblems, filtering out those that fail verification by generating and verifying repeatedly until passing, within a maximum retry limit.
>
> ```Python
> from math_verify import parse, verify
> if answer:
>     try:
>         parsed_answer = parse(answer)
>         direct_answer_prompt = f"""
> Solve this question step by step.
>
> {question}
>
> Provide only the final numerical answer (e.g., 7.5, 4, 5/324), no explanation.
> """
>         direct_answer = call_gpt(client, deployment_name, direct_answer_prompt).strip()
>
>         parsed_direct_answer = parse(direct_answer)
>         is_verified = verify(parsed_answer, parsed_direct_answer)
> ```
> ---
> We thank the reviewer again for the constructive comments. We would be more than happy to further discuss if there are any remaining concerns or questions!

---

> > ### Comment · Reviewer_A4Mp · 2025-08-06
> >
> > Thank you for the detailed responses. I’m satisfied with the clarifications, and most of my concerns have been addressed. I will raise my score accordingly.
> >
> > Regarding W1, I would like to clarify that by "long CoTs", I was specifically referring to model-generated responses that include extended thinking processes—such as the long-form reasoning traces produced by models like R1 in the DeepMath-103K dataset. It would strengthen the paper if the authors could elaborate on how their method handles such cases, possibly with examples or a brief analysis.

---

> ### Author Response · Authors · 2025-08-08
>
> We sincerely thank the reviewer for acknowledging our detailed rebuttal and for the score increase based on our clarifications. We address the remaining concern below:
>
> Thank you for the clarification. After carefully looking into the DeepMath-103K dataset, we now understand that your concern is about how our method handles cases where the original data contains long CoTs generated by reasoning models such as DeepSeek R1, which often include noisy information such as dead ends and redundant backtracking. This is a valuable question, and we have in fact already conducted experiments on datasets with very long CoTs. We appreciate the opportunity to highlight these results here in direct response to your concern.
>
> We would like to highlight the experimental results we presented during the rebuttal phase in response to reviewers eiT7 and UED6, which were positively acknowledged by both reviewers. These experiments were conducted in a more general domain, competitive programming, where long and complex reasoning traces are common. The dataset used was Codeforces-CoTs from HuggingFace Open-R1, which contains solutions generated by DeepSeek-R1 and exhibits the characteristics you described. Our method achieved substantial improvements over baselines on both the Qwen2.5-1.5B-Instruct model and the DeepSeek-R1-Distill-Qwen-1.5B model, as shown below, demonstrating the effectiveness of our approach on long CoT solutions.
>
> | Model | HumanEval pass@1 |
> | --- | --- |
> | Qwen2.5-1.5B-Instruct | 34.15 ± 3.71 |
> | Qwen2.5-1.5B-Instruct-SFT | 35.37 ± 3.74 |
> | Qwen2.5-1.5B-Instruct-DecompX (Ours) | **42.68 ± 3.87** |
> | DeepSeek-R1-Distill-Qwen-1.5B | 36.01 ± 1.85 |
> | DeepSeek-R1-Distill-Qwen-1.5B-DecompX (Ours) | **57.90 ± 0.98** |
>
> ---
>
> To directly address the reviewer’s request for an example and brief analysis, we present the following case illustrating how our method handles such long CoT solutions. Specifically, consider the Codeforces problem Another n-dimensional chocolate bar (ID 1801F, row 42 in the dataset), whose CoT spans 17,886 tokens. The original trace contains extensive meta-cognitive commentary such as “Wait”, “Maybe”, and “Another idea…”, exploratory dead ends, and repeated backtracking.
>
> Since the complete example far exceeds the response character limit, we present here only a partial illustration. The full decomposed data will be included in the final version. In the table below, the left column paraphrases the original CoT steps, while the right column presents the concrete steps and tags extracted by our method.
>
> | Original CoT (paraphrased) | Tags and Reasoning Step in DecompX |
> | --- | --- |
> | “If we multiply all `a[i]` and it is still < k then… wait, right, then answer must be 0. We can early stop when the product already ≥ k.” | **Early Exit on Infeasibility**: compute `A_prod` with early stop; if final `< k`, return 0. |
> | “To increase `∏ b[i]` with the least damage, we should pick the smallest `(b+1)/b`… hmm that happens when `b` is largest… so use a max-heap on `b[i]` and increment that index.” | **Priority Queue**: while `B_prod < k`, pop index with largest `b[i]`, do `b[i]+=1`; update `B_prod`. |
> | “We can jump instead of plus-one many times: compute the exact `t` that just passes k, set `b[i]=t`.” | **Batched Jump**: `t = min(a[i], ceil(k * b[i] / B_prod))`; update `B_prod` and `b[i]`. |
>
> Despite the extensive and sometimes noisy narrative, our method extracts a clean sequence of code-level reasoning steps such as early infeasibility checks, heap-based greedy growth with optional batching, interval sanity checks, and numerically safe final computations.
>
> ---
>
> We note that the Codeforces-CoTs dataset is in fact longer and structurally more complex than DeepMath-103K, making it a highly representative choice for evaluating long CoT handling. The statistics are as follows:
>
> **DeepMath-103K (reviewer’s example):**
> * Average tokens per solution: 5,230
> * Min: 894 tokens
> * Max: 23,401 tokens
>
> **Codeforces-CoTs (used in our experiments):**
> * Average tokens per solution: 12,839
> * Min: 980 tokens
> * Max: 24,043 tokens
>
> Given these statistics, we believe our results demonstrate the method’s capability in handling long CoTs. Nonetheless, we greatly appreciate the reviewer's suggestion of this very recently released in April dataset and happy to incorporate experiments on DeepMath-103K in the final version as a dedicated case study for the math domain.
>
> ----
>
> Once again, we thank the reviewer for raising this valuable point, which has allowed us to further showcase the strengths of our approach in long-CoT scenarios. We would be more than happy to further discuss if there are any remaining concerns or questions!

---

### Comment · Area_Chair_ehor · 2025-08-04
**Engage in discussion**

Dear reviewers A4Mp and eiT7,

Thank you for your thoughtful feedback! As the rebuttal comments are now available, we kindly encourage you to read the other reviews and the authors’ response carefully. If you have any follow-up questions, please raise them soon, so the authors can respond in a timely manner.

AC

---

### Author Response · Authors · 2025-08-09
**Global Response**

We sincerely thank all reviewers for their thoughtful and constructive feedback, and for their recognition of our method’s novelty, clear writing, solid experimental results, and promising direction, as well as their acknowledgment of our detailed rebuttal clarifications.

During the rebuttal and discussion phase, we have addressed all raised concerns and questions as follows:

* **Handling long CoTs and scaling (R-A4Mp)**: We clarified and demonstrated strong performance on datasets with extremely long, noisy reasoning traces (e.g., Codeforces-CoTs, avg. 12.8k tokens/solution), showing our framework extracts clean, verifiable steps even under such conditions.

* **Small training sets and fair comparison (R-A4Mp, R-UED6)**: All methods start from identical seed data; we normalize compute budgets via token-count–adjusted epochs, ensuring gains stem from structured curricula, not more seed data.

*  **Baseline breadth and competitive comparisons (R-A4Mp, R-UED6, R-nBar)**: We added results against direct distillation, MuggleMath, and the competitive public baseline, showing consistent gains. Our method also enhances reasoning-capable models (e.g., DeepSeek-R1-Distill-Qwen-1.5B).

* **Generalization beyond math (R-eiT7, R-UED6, R-nBar)**: We demonstrated cross-domain, cross-language, and out-of-distribution benefits, including MATH→AIME2025, AIME2024→AIME2025, and Codeforces→HumanEval.

* **Teacher model dependence (R-UED6, R-nBar)**: While current results use strong teacher models and small student models, our design includes verification/retry to mitigate noise and naturally supports future “weak-to-strong” and self-improvement scenarios.

We also incorporated reviewer suggestions to discuss educational applications (e.g., automatic scaffolding, dynamic curriculum design), broader applicability to structured reasoning domains, and potential future work combining decomposition with large-scale distillation.

We believe these clarifications and additional experiments address the raised concerns and further strengthen the case for the significance, generality, and robustness of our proposed method.

---

### Note · Authors · 2025-08-16

We sincerely thank all reviewers for their careful reviews and constructive discussions. We are encouraged that several reviewers raised their scores after our clarifications and additional experiments.

**Reviewer Highlights:**
- **A4Mp:** “The core idea is novel,” “better generalization on OOD,” “a principled way.” After clarifications: “satisfied.”
- **eiT7:** “Well written,” “clear and sound,” “supported by solid results.”
- **UED6:** “Clear and convincing,” raised score after clarification.
- **nBar:** “Novel methodology,” “dataset insights,” “step in an interesting direction,” recognized added experiments showing advantages over direct distillation.

**Key Clarifications & Results:**
1. **Long CoTs:** On CodeForces-CoTs (avg. 12.8k tokens, noisier than DeepMath-103K), DecompX successfully extracted clean steps and improved performance, showing robustness on long reasoning traces.
2. **Reasoning Students:** Fine-tuning **DeepSeek-R1-Distill-Qwen-1.5B** with DecompX yielded large gains, confirming benefits for reasoning-capable students.
3. **Beyond Math:** In coding (**CodeForces → HumanEval**), DecompX improved pass@1 for both Qwen2.5-1.5B-Instruct and DeepSeek-R1-Distill-Qwen-1.5B, showing cross-domain and cross-language generalization.
4. **Baselines:** We added **Direct Distillation** and **MuggleMath**, under equal compute/token budgets, confirming consistent improvements.
5. **OOD Generalization:** Strong gains across **AIME2024→AIME2025**, **MATH→AIME2025**, and **CodeForces→HumanEval**, highlighting robustness across tasks and domains.
6. **Clarifications:** We detailed prompts, verification ensuring solvable subproblems, and training/test splits. We also analyzed failure cases and discussed limitations.
7. **Beyond Distillation:** While current work uses a “large teacher → small student,” our design naturally supports **weak-to-strong generalization** and **self-improvement**, where models of similar capacity generate structured curricula for peers.

**Impact:**
DecompX offers a principled, interpretable, and cost-efficient way to turn limited supervision into structured curricula, improving both base and reasoning-capable models while enhancing OOD robustness. We believe this establishes a foundation for recursive self-improvement and has broad potential impact.

We again thank the reviewers and ACs for their valuable input, which we will integrate into the final version.

---

### Decision · Program_Chairs · 2025-09-17

**Decision:**

Accept (poster)

**Comment:**

This paper received three borderline accept and one accept rating. The authors raised some concerns, which were mostly addressed in the rebuttal by authors. The reviewers appreciate, the novelty of the method, the clear writing, and the insightful discussion. The AC agrees that the paper should be accepted.